# Adaptive elastic convolution-based YOLO for peripheral blood smear cell detection

Neha Margret Issac[1☉], Rajakumar K.[2☉*]

1 School of Computer Science and Engineering, Department of Analytics, Vellore Institute of Technology, Vellore, Tamilnadu, India, 2 School of Computer Science and Engineering, Department of Analytics, Vellore Institute of Technology, Vellore, Tamilnadu, India

☉ These authors contributed equally to this work.
* rajakumar.krishnan@vit.ac.in

## Abstract

Peripheral blood smear analysis continues to be important for the diagnosis of various hematologic conditions, although automated systems often experience difficulties due to the variety of cell types and the multi-class detection requirements associated with them. In this work, we propose Elastic YOLO (EYOLO), an extension of the You Only Look Once (YOLO) object detection framework designed for morphology-aware detection of red blood cells, white blood cells, and platelets in peripheral blood smear images. In a dataset annotated by clinicians, Elastic YOLO achieved a mAP@0.5 of 94.7% and mAP@0.5:0.95 of 87.8%, outperforming the baseline YOLOv5 model and several recent detection architectures while achieving inference speeds of up to 78 frames per second (FPS) under high-performance GPU settings (NVIDIA RTX 4090) at $256 \times 256$ input resolution. Elastic adaptive convolutions form the core of the framework, allowing the receptive field to adapt dynamically to variations in cell size, shape, and staining conditions across different cell types. The automated smear image analysis offered by this system may assist computer-assisted hematology workflows and can support remote screening scenarios such as telemedicine-based triage, where rapid preliminary analysis of peripheral smear images is required.

## Introduction

Peripheral blood smear (PBS) examination is a fundamental diagnostic tool used in clinical hematology to assess the morphology of blood cells and identify pathological abnormalities [1]. PBS analysis plays an essential role in diagnosing a wide range of hematological disorders, including anemia, infections, leukemia, and platelet dysfunction [2]. Despite its clinical importance, smear examination is traditionally performed manually under a microscope, which is time-consuming and highly dependent on expert interpretation [3].

In modern laboratories, automated hematology analyzers perform initial screening through complete blood count (CBC) measurements [4,5]. While these systems

**Data availability statement:** The dataset for normal blood cell detection (TXL-PBC) is publicly available at: https://github.com/lugan113/TXL-PBC_Dataset/tree/master/TXL-PBC with DOI: https://doi.org/10.1038/s41597-025-05980-z. The curated abnormal peripheral blood smear dataset (PBSS-Abnormal Dataset), derived in part from images obtained from the ASH Image Bank (https://imagebank.hematology.org/), is publicly available via Zenodo: https://doi.org/10.5281/zenodo.19354729. This dataset contains abnormal peripheral blood smear images categorized into abnormal promyelocytes (APL), abnormal red blood cells (ARBC), and abnormal white blood cells (AWBC). The current version contains image data only; expert-validated annotations will be released in a future version. Additional evaluation images were obtained from the ASH Image Bank (American Society of Hematology) and are used in accordance with their licensing policies.

**Funding:** The author(s) received no specific funding for this work.

**Competing interests:** The authors have declared that no competing interests exist.

provide reliable quantitative parameters, they often struggle to accurately identify morphological abnormalities or rare cell types. Consequently, flagged samples still require manual PBS examination by trained specialists [6]. This manual process introduces workload burden and inter-observer variability, particularly when subtle morphological variants or rare abnormal cells are present [1,2].

Various physiological factors and disease processes contribute to the intricacy of the problem for automated systems to recognize blood cells, which exhibit greater variation in morphology [7]. On the other hand, there are blood cells that are consistently present and can aid in diagnosing diseases and infections. In early-stage malaria infection, red blood cells (RBC) tend to maintain their normal morphology. In addition, abnormal lymphocytes tend to exhibit greater variation in morphology compared to normal lymphocytes in various viral and reactive conditions, characterized by an increase in size, irregularities in nuclear outline, and an abundant amount of basophilic cytoplasm [8]. These abnormal cells are difficult to recognize by automated systems due to their similarity to other abnormal white blood cells and variation in presentation [7,9,10].

Machine learning-based techniques for PBS analysis have shifted from traditional handcrafted machine learning techniques to fully automated convolutional neural network (CNN) techniques [10,11]. This has been driven by advances in computer vision and deep learning techniques (DL). You Only Look Once (YOLO) techniques [12–15], which are known for their efficiency in real-time object detection, are highly promising for AI-based hematology and tele-diagnostic techniques.

Even with these advancements, fine-grained morphological classification is still difficult. In addition, many existing YOLO- and CNN-based PBS analysis systems encounter practical limitations in real laboratory conditions. Overlapping cells, subtle morphological variants, and variations in staining protocols or microscope acquisition settings can introduce domain shifts between datasets, which often reduce the robustness of models trained on limited or homogeneous image sources. RBC variations like target cells, spherocytes, and elliptocytes [9,16–18], and even rare WBC types such as leukemic blasts [19–23] are often misclassified because of high similarities within classes. The presence of platelet clumping or abnormal/atypical granules increases the difficulties in the automated detection process. Only a few systems truly consider the morphological invariants explicitly [6,24–26]. Architectures based on the Transformer increase the representation of features, but they are not cost-effective and thus less appropriate for applications that are low on resources or that are real-time.

Elastic YOLO is presented as an enhanced version of the YOLO framework, which is adapted to overcome the shortcomings in blood cell detection technology, and it incorporates elastic modules in both the backbone and neck parts. The dynamic control of the receptive field by these modules enables the extraction of adaptive features from cells that exhibit different morphological patterns and staining properties. The entire concept provides a design that effectively recognizes the morphological invariants and variants among white blood cells, red blood cells, and platelets, offering the benefits of stability and real-time processing performance.

In contrast to previous approaches that have considered deformable convolution and architectural scaling as independent extensions, Elastic YOLO unifies spatial

elasticity and architectural elasticity under a unified object detection framework. The deformable convolution technique is used to vary the receptive field geometry, thereby capturing small morphological changes, while dynamic depth-width scaling adjusts the capacity of the model with increasing cell complexity. The coupling of these two forms of elasticity allows Elastic YOLO to adapt both what features are extracted and how much model capacity is allocated, rather than relying on fixed design-time assumptions. To the best of our knowledge, this is the first YOLO-based hematology detection framework to explicitly unify spatial, architectural, and computational elasticity for morphology-aware peripheral blood smear analysis.

The main contributions that have been made in this paper are listed below:

- An EYOLO or elastic YOLO architecture capable of detecting blood cells, including RBCs, WBCs, and platelets, which are normal, morphologically invariant, and abnormal types.

- The use of adaptive elastic convolutional blocks that will help to reduce intra-class similarity and increase inter-class discriminability.

- The dataset used for the evaluation was a clinically validated PBS dataset, and the results showed that the proposed method demonstrates improved performance over the standard YOLO and state-of-the-art detection frameworks.

- A real-time detection system suitable for hematology analyzers, point-of-care diagnostics, and telemedicine-based platforms has been developed.

## Related work

### Evolution of blood smear analysis techniques

The PBS analysis has shifted from traditional manual microscopy to diagnostic systems using deep learning. The traditional image processing techniques, such as Otsu thresholding, Watershed, and Contour, were able to detect features, but these techniques were not very successful when the staining, overlapping, and shape variations were significant [6].

Traditionally, many image analysis techniques were feature-based. The image analysis techniques used feature detectors like local binary patterns (LBP), histogram of oriented gradients (HOG), and scale invariant Feature transform (SIFT), and traditional classifiers like Random Forest and SVM. Although these techniques were accurate, the need for human involvement in feature tuning and parameter selection was high [4,27].

With the advent of deep learning, CNNs like ResNet, U-Net, and VGGNet were successful in automating blood cell classification and segmentation effectively [5,18].

To briefly describe, DL is a set of neural networks that use multiple layers of abstraction to learn complex features from the image. Darknet is an open-source neural network library originally created for YOLO's earlier versions, while PyTorch is an open-source library used for developing, training, and deploying neural networks. Vision Transformers are an extension of the transformer architecture, which is originally used in natural language processing. Affine transformations are geometric transformations that preserve the relative position of points in an image.

The YOLO family of DL is significant in the field of hematological image analysis due to its real-time capability and detection accuracy. Although earlier versions, like YOLOv3 and YOLOv4 [21], were successful, these were limited due to their dependence on the Darknet library. The recent introduction of YOLOv5, which is built on PyTorch, has made the customization, training, and deployment process much easier across heterogeneous computing environments.

### YOLOv5–based studies

The YOLOv5 model has been a preferred choice for various hematology studies owing to the good trade-off between the speed and computational requirements of the model [6,7]. In their studies, Ferreira and Couto [2] employed the YOLOv5

model for the detection of white blood cells, focusing on the diagnosis of leukemia. Tarimo et al. [12] also attempted the integration of YOLOv5 and Vision Transformers for the improvement of the precision of the white blood cell classification. In another study, Bai et al. [18] employed the YOLOv5 and U-Net++ for the detection and segmentation of the white blood cells through a two-step approach. In a recent study, He et al. [19] introduced optimizations for the YOLOv5 model, and later, Shams [28] and Naing [22] demonstrated the robustness of the YOLOv5 model for various conditions.

## Advancements in YOLOv8 and its derived architectures

YOLOv8 is the next-generation object detection framework that has several high-level changes in its architecture, including C2f modules, anchor-free detection heads, and better label assignment strategies. In the field of hematology, Abozeid *et al.* [1] employed YOLOv8 for WBCs detection, while Luong *et al.* [16] developed the framework further to distinguish WBCs, platelets, and RBCs but this came with the price of greater computational demands. Different derivative models have been put forward: YOLO-FMS [29] is meant for detecting small cellular structures, Gpmb-YOLO [13] grabs computational efficiency as its natural core, SSW-YOLO [7] fits Swin Transformer modules for accurate feature extraction, and DWS-YOLO [15] is made for portable implementations. Apart from that, in the same line with YOLOv9 [10], excellent results were achieved in the detection of malaria-infected cells. The same goes for the hybrid approaches of YOLO-CNN [11], which were aimed at the reticulocyte classification by means of the fused convolutional representations.

## Transformer-integrated & morphology-aware architectures

Transformer-augmented YOLO architectures, such as WBC YOLO-ViT [12] and the model proposed by Wu *et al.* [9], are very effective in recognizing different cell shapes by capturing long-range contextual dependencies, thus becoming one of the reasons for the modern-day cell morphologies' recognition improvement. These architectures, however, despite the better generalization, come with a higher computational cost. Other proposed changes in the architecture, e.g., large-kernel convolutions [8] and guided-attention use [27], increase the immunity of the system to staining artifacts and lighting variations further.

## Identified challenges in current research

Some of the significant gaps identified in the literature are:

1. **Absence of an all-inclusive framework for the detection of multi-class blood cells:** The present-day models mainly focus on the classification of single cells or highlight certain morphological features [1,8,12].

2. **Rigid Architectural Design:** A lot of models are based on the pre-defined complexity, which reduces their capability to be flexible with different computing/hardware conditions [2,7,13,16,19,29].

3. **High Resource Requirement:** YOLOv8 [1,16], and transformer-based YOLO variants [9,12] are among the models that need huge GPU computation for both training and inference.

4. **Data-Driven Method:** Multiple methodologies rely on large labeled datasets [2,12,18].

5. **Morphological Sensitivity:** Variability in light, staining, and slide preparation continues to introduce errors in misclassification [7–9,27].

## Proposed research goals and strategic objectives

The proposed method, Elastic YOLO, has the following features to address the above drawbacks:

• Complete recognition of RBCs, WBCs, and platelets, together with their different forms.

- Real-time changing of model depth, width, and input resolution for optimal performance.

- Lightweight high-precision deployment that is fit for embedded and low-power systems.

- Application of complex augmentation and transfer learning methods for efficient data training.

- Use of elastic scaling and deformable convolution layers to learn morphology-invariant representations for better generalization across the diversity of cells.

This model has been placed between the rigid and high-capacity variants like YOLOv5 and YOLOv8, and the lighter variants like YOLO-FMS and Gpmb-YOLO. It has been designed to provide a mix of flexibility, efficiency, and diagnostic accuracy.

## Proposed methodology

The **Elastic YOLO** framework has been designed to automatically identify and classify the components of a peripheral blood smear (PBS) image. It detects normal white blood cells (WBC), platelets (PL), and red blood cells (RBC), and abnormal variants of RBC, AWBC, and APL.

Further, it has been made possible by the adaptive scale of the YOLO family, which makes it capable of dealing with the kind of variation that often occurs in blood sample analysis. Six steps are involved in this workflow, as depicted in Fig 1.

## Dataset overview

The dataset used in this study consists of peripheral blood smear images collected from multiple sources to ensure variability in staining conditions, imaging devices, and cell morphology. The dataset contains a total of 4,486 images annotated with six cell classes:

$${RBC, WBC, PL, ARBC, AWBC, APL}$$

where RBC, WBC, and PL represent normal red blood cells, white blood cells, and platelets, while ARBC, AWBC, and APL represent abnormal variants.

Images were obtained from the publicly available TXL-PBC dataset, the ASH Image Bank, and laboratory-generated smear samples. All abnormal cell annotations were verified by experienced pathologists to ensure clinical reliability.

## Dataset, annotations, and preprocessing

In the present study, the images used from the PBS image dataset are from two different sources, ensuring the images are varied and reliable from a medical point of view. For the images of normal red blood cells, white blood cells, and platelets, the data used were from the TXL-PBC image dataset [30]. In addition, images of abnormal and infected blood cells were taken from lab-generated images and the ASH Image Bank [31], a source of images verified for medical accuracy, which provides images of various hematology-related problems. Only publicly available datasets and lab-generated images, approved and obtained through ethical means and medical practice, are used for the present study.

The abnormal categories, namely abnormal RBC (ARBC), abnormal WBC (AWBC), and abnormal platelets (APL), correspond to different pathological conditions like parasitic infections, morphological deformities, and disease-induced cytological variations. To ensure class balance for the abnormal cell images, a few synthetic images of abnormal blood cells were added, ensuring the accuracy of the cytology of the images. For this purpose, a small set of synthetic images of abnormal blood cells was added, and the Adaptive Generative Adversarial Network (A-GAN) [32] was used for generating the images. However, the use of A-GAN for image generation and the addition of the images for class balancing and

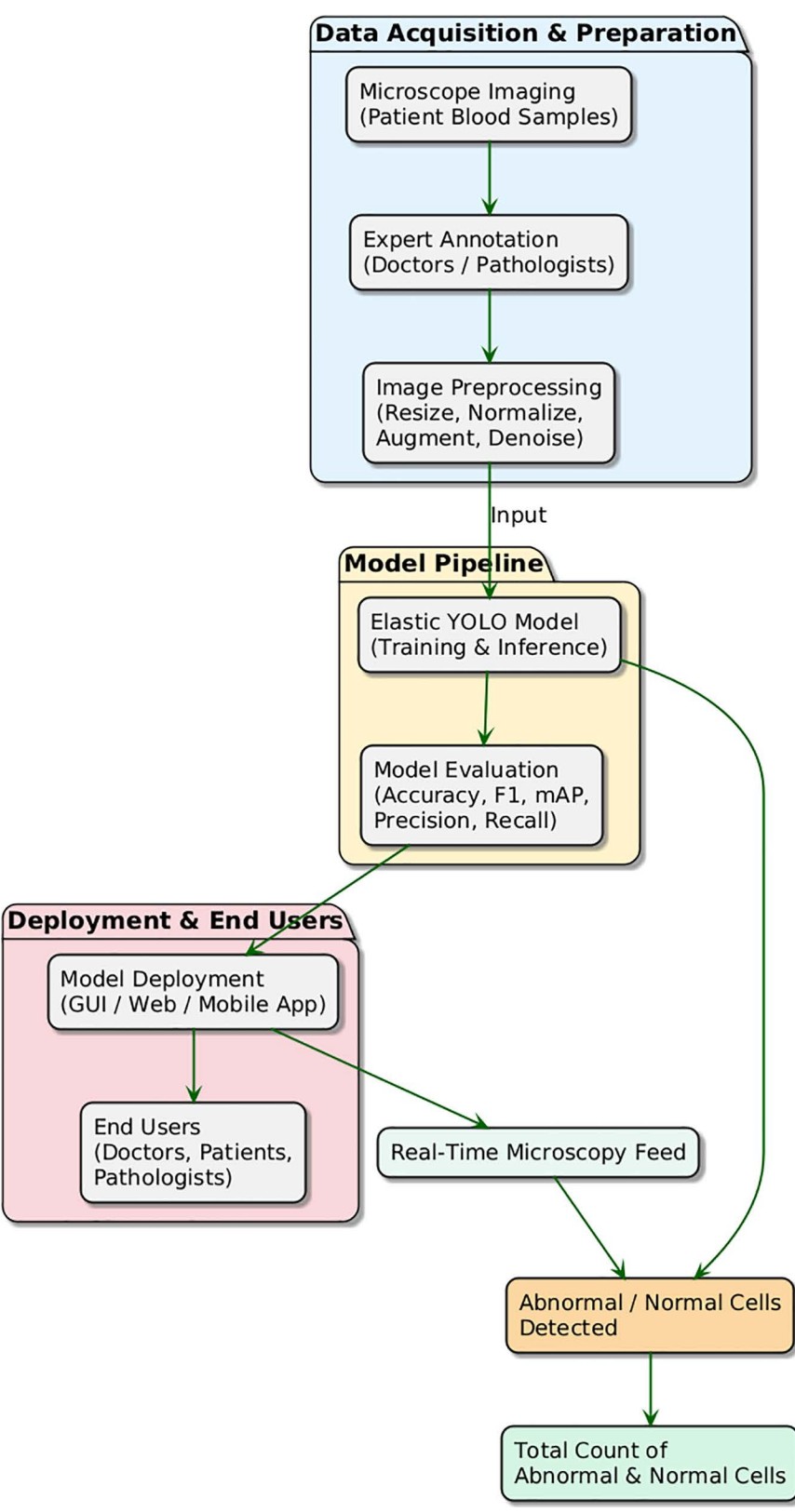

**Fig 1. Comprehensive workflow of the Elastic YOLO detection system.**

ensuring the accuracy of the cytology of the images are done without affecting the performance metrics of the system, and the use of A-GAN does not affect the evaluation of the quantitative performance of the system. In fact, the quantitative performance metrics of the system are calculated only on the basis of the images from the datasets and are not affected by the use of the A-GAN for image generation and the addition of the images for the purpose of class balancing and ensuring the accuracy of the cytology of the images.

In total, the final dataset is made up of **4,486 images** labeled with six cell classes (samples from the dataset are shown in Fig 2):

$$\{RBC, WBC, PL, ARBC, AWBC, APL\}.$$

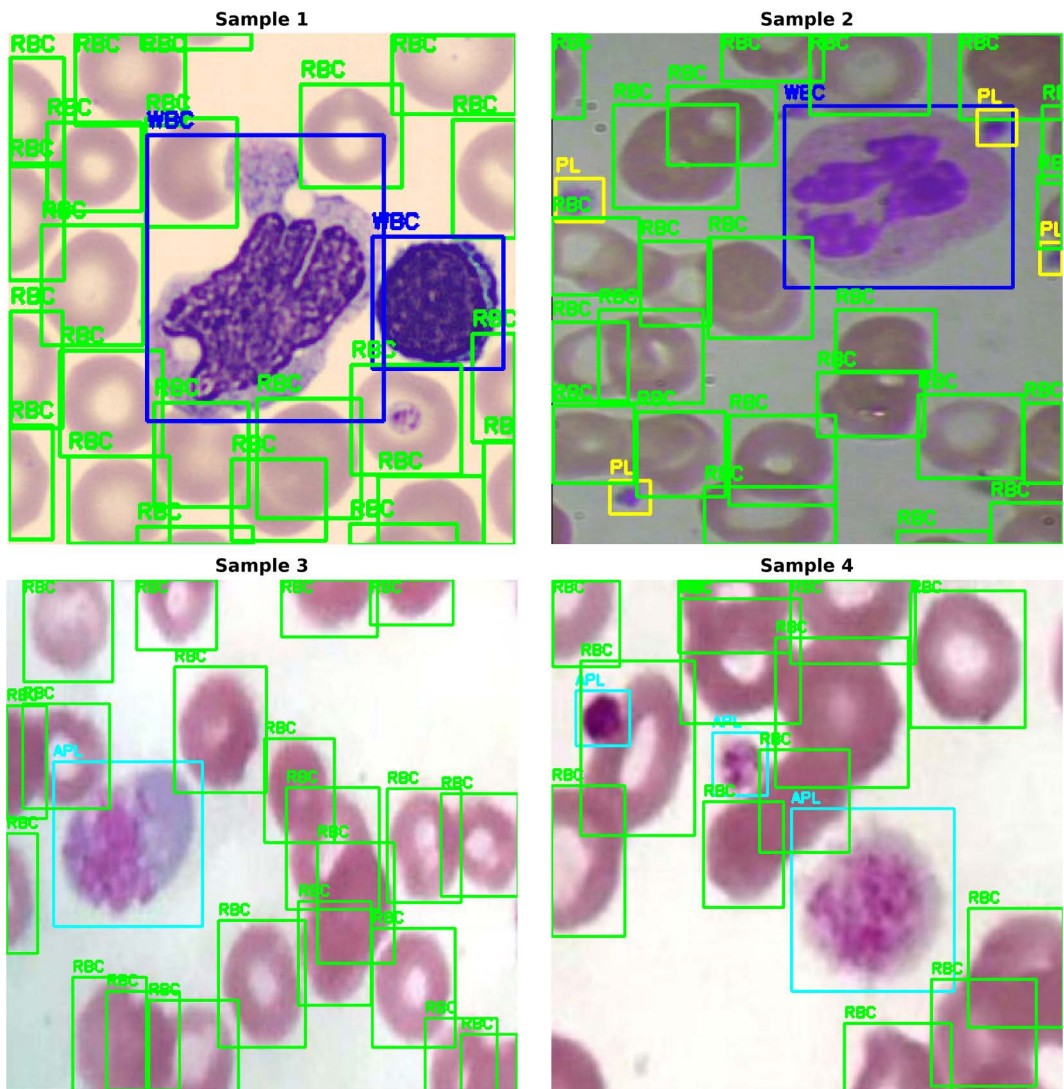

**Fig 2. Representative ground-truth annotation samples from the curated peripheral blood smear dataset.**

The addition of various data sources, staining techniques, and acquisition conditions was aimed at simulating the reality of a heterogeneous laboratory environment rather than a controlled single imaging scenario.

**Expert Annotation and Validation:** All the annotations of abnormal and infected cells, which were derived from the laboratory samples and ASH Image Bank [31], were thoroughly checked and verified by professional medical pathologists. By doing this, it was ensured that the abnormal annotations were exactly the same as the true pathological features and not related to any stain used in the process or any noise in the image. In addition, a senior lab technician also verified some of the annotated images to minimize the difference between the annotators and thereby maintain inter-annotator consistency.

The abnormal white blood cell class used in this study includes a variety of abnormal and atypical white blood cell types, including reactive lymphocytes, blast-like cells, and dysplastic forms that occur under pathological conditions. Although there is a lack of information regarding specific white blood cell types, including basophils and monocytes, which might be considered as individual classes, abnormal white blood cell classes reflect those abnormalities that commonly occur with analyzer flags. The objective of this study is not to classify each white blood cell individually, but to accurately localize abnormal and normal cell morphology under conditions that commonly cause difficulties with automated hematology analyzers.

**Dataset Structure:** The table 1 below illustrates the number of images and annotations used for each class. The dataset is highly dominated by a high number of annotations used for RBCs because RBCs are the most commonly occurring cell types when working with blood smear images. Although there is a high number of RBCs, abnormal white blood cell classes are also important because, although there is only a small number of abnormal samples, they are critical for use.

**Train–Validation–Test Split:** To get an unbiased evaluation, the dataset was separated with a standard subject-independent split: 70% was used for training (3,140 images), 15% for validation (673 images), and 15% for testing (673 images). The same acquisition batch images were not split across partitions, thus eliminating data leakage and enhancing the assessment of generalization.

**Ethical Considerations:** All data used in this study were obtained from publicly available datasets and ethically approved laboratory sources. The ASH Image Bank data were used in accordance with their licensing policies. No patient-identifiable information is included in this study.

**Image Preprocessing and Augmentation:** All the images were resized to a fixed width and height of $256 \times 256$ pixels. The pixel values of each image were then normalized with respect to the mean and the variance of the pixel grey levels, as calculated within the training procedure:

$$I_{norm}(x, y) = \frac{I_{orig}(x, y) - \mu}{\sigma},$$

(1)

**Table 1. Dataset Composition and Annotation Statistics.**

| Class | Images | Annotations | Source |
|---|---|---|---|
| RBC | 1120 | 8945 | TXL, Lab |
| WBC | 780 | 1560 | TXL, Lab |
| PL | 526 | 2104 | TXL |
| ARBC | 620 | 4312 | ASH, Lab, GAN |
| AWBC | 540 | 1985 | ASH, Lab, GAN |
| APL | 400 | 1268 | ASH, GAN |
| **Total** | **4486** | **20174** | – |

TXL denotes the TXL-PBC dataset; ASH refers to the ASH Image Bank; Lab indicates laboratory-acquired images; GAN represents synthetically generated abnormal samples.

where $\mu$ and $\sigma$ are the mean and standard deviation of the training images, respectively.

To maintain the diagnostic morphology and, at the same time, to enhance the robustness, only mild affine transformations were performed in the course of training:

$$I_{\text{aug}} = T(I_{\text{norm}}, \theta, s, t_x, t_y),$$

(2)

In the mentioned range, $\theta \in [-15°, 15°]$ stands for rotation, $s \in [0.9, 1.1]$ stands for scaling, and $t_x, t_y \in [-0.1R, 0.1R]$ indicate translations. The application of color jittering, elastic deformation, or intensity distortion was completely avoided because they would have an impact on the clinically important cell morphology.

**Statistical Description of Annotations:** The statistical distribution of annotation density per image is summarized in Table 2, highlighting the inherent class imbalance between normal and abnormal samples. Normally, the images will have more objects due to the high count of red blood cells. Abnormal images, however, will have fewer objects but are more diagnostically important. The statistics indicate the inherent class imbalance in practical hematology data and the need for the hybrid detection and anomaly analysis strategy that was used.

## YOLOv5 baseline framework

The baseline model that was used is built on the YOLOv5 detection pipeline, and it includes three core components:

1. a Backbone of CSPDarknet for hierarchical feature learning/extraction, 2. a feature pyramid network (FPN) for multi-scale feature fusion, 3. a path aggregation network (PANet) for forward and backward propagation of features.

Such a combination allows for the detection of small (platelets), medium (lymphocytes, ARBCs), and large (monocytes, giant RBCs) cellular structures at the same time. Fig 3 illustrates the conventional YOLOv5 workflow.

## Elastic YOLO architecture adaptation

Unlike static YOLO variants, the Elastic YOLO architecture introduces adaptive control over both depth and width dimensions.

Let $\lambda_d$ and $\lambda_w$ represent depth and width scaling parameters, respectively, where $\lambda_d \in 1.25, 1.0, 0.75$ and $\lambda_w \in 1.5, 1.0, 0.5$. These values dynamically adjust the number of channels & layers per stage to suit the complexity of the input morphology, enhancing generalization for a variety of blood cell types.

The elastic convolution operator is the key component of the current network. Instead of using a predetermined channel count, the number of output channels is adaptively determined at run-time.

This operation at the $j$-th layer can be written as:

$$\Phi_j : Z^{(j)} \in \mathbb{R}^{H_j \times W_j \times D_j} \rightarrow Z^{(j+1)} \in \mathbb{R}^{H_{j+1} \times W_{j+1} \times D_{j+1}^{\text{out}}}$$

(3)

**Table 2. Descriptive Distribution of Annotations Per Image.**

| Metric | Normal | Abnormal | Overall |
|---|---|---|---|
| Mean | 7.9 | 3.4 | 11.3 |
| Std. Dev | 2.8 | 1.9 | 3.6 |
| Min | 2 | 0 | 3 |
| Max | 27 | 14 | 38 |
| Median | 8 | 3 | 12 |

A statistical summary has been created which describes how the number of normal and abnormal cell annotations varies between images in the entire dataset.

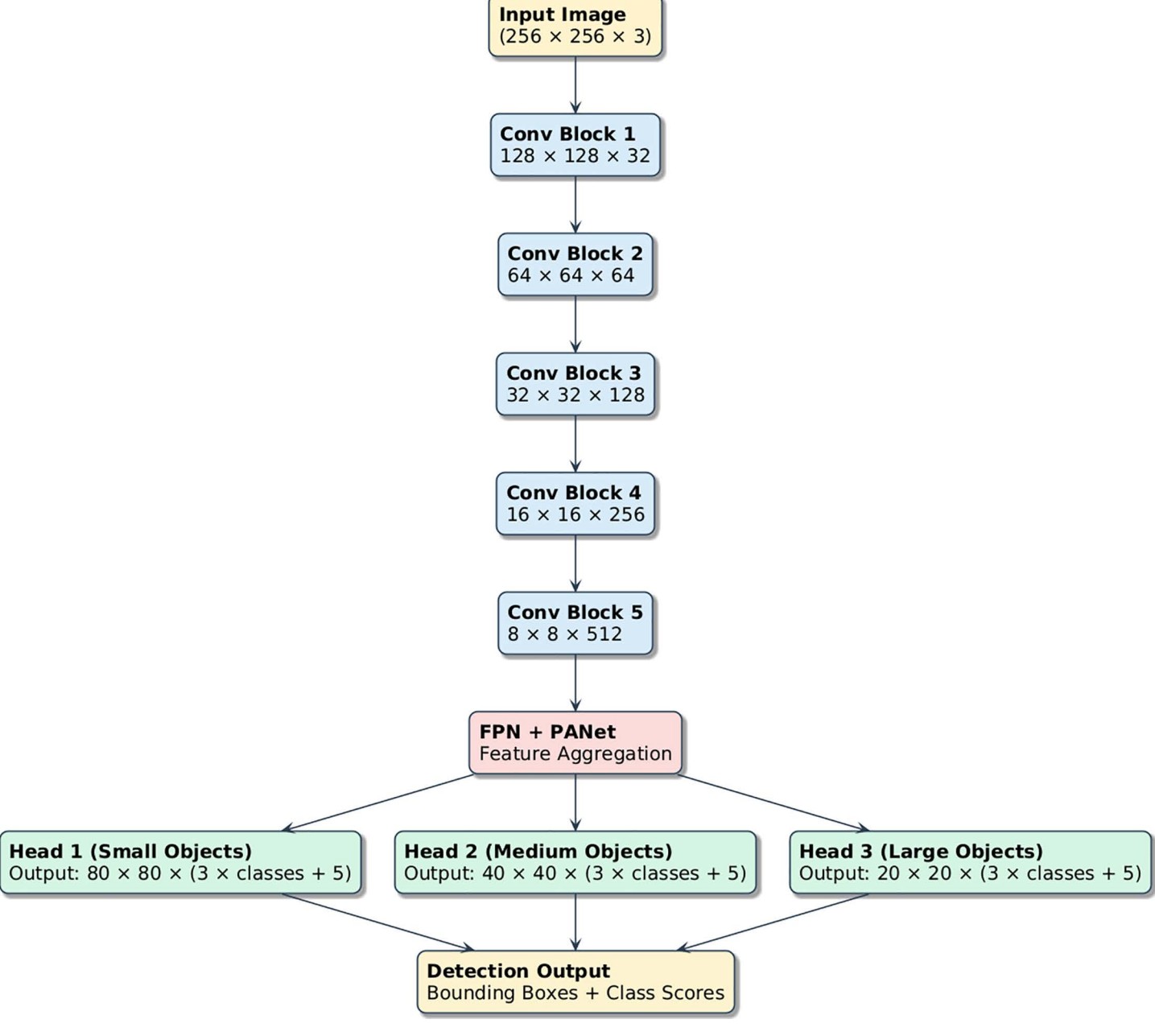

**Fig 3. Multi-scale blood cell detection using YOLOv5 architecture.**

where $W_j$, $H_j$, and $D_j$ represent the width, height, and input channels, while $D_{j+1}^{out}$ is the adaptively determined number of the output channels in the range $D_{j+1}^{out} \in [D_{j+1}^{min}, D_{j+1}^{max}]$.

Spatially adaptive feature learning is achieved through deformable convolution, described as:

$$f(\xi_0) = \sum_{\xi \in \mathcal{S}} \omega(\xi); g(\xi_0 + \xi + \Delta\xi)$$

(4)

where:

- $\xi_0$ – the reference spatial coordinate that is on the output map,

- $\mathcal{S}$ – the area of the convolution kernel where sampling takes place,

- $\xi$ – the base offset applicable for every sampling position,

- $\Delta\xi$ – the vector of displacement that can be learned and which adds geometric flexibility,

- $\omega(\xi)$ – the weights of the kernel corresponding to each offset $\xi$,

- $g(\cdot)$ – the activation of the input feature.

The elastic feature pyramid networks (FPN) and Path Aggregation Network (PANet) will retain essential connections and reconfigure the feature flow paths dynamically, and will reduce floating-point operations (FLOPs) and maintain detection accuracy for the elastic YOLO.

The whole concept of the adaptive elastic YOLO has been clearly depicted in Fig 4.

## Architectural comparisons from a comparative perspective

- The Dynamic Scalability of Elastic YOLO allows for real-time adjustment of not only the number of layers but also the number of channels at any given layer, whereas YOLOv5 requires knowledge of how many channels to add and/or remove before the model starts.

- The Receptive Field Management capability of Elastic YOLO use Deformable Kernel to increase the amount of information that the kernel can sample for each pixel, allowing it to model the morphological relationship between structures more accurately.

- Selective Pruning reduces Latency for inference through the cessation of unneeded feature propagation within the Elastic YOLO.

- Elastic YOLO maintains the ability to provide consistent and accurate detection regardless of differences in the geometric forms of cells or the presence of Staining artifacts or differences in magnification.

## Detection logic and visualization

The network generates a collection of bounding boxes $\mathcal{B}$ accompanied by corresponding confidence scores, as defined in:

$$\mathcal{B} = \{(\mathbf{b}_t, \mathbf{p}_t) \mid t = 1, \ldots, T\} \tag{5}$$

here, $\mathbf{p}_t \in \mathbb{R}^C$ denotes the class-confidence vector that corresponds to $C = 6$ categories, while $\mathbf{b}_t = (x_t, y_t, w_t, h_t)$ denotes the bounding box pixel coordinates.

The class predictively assigned $\hat{\lambda}_t$ is computed as follows:

$$\hat{\lambda}_t = \arg \max_{c \in \{1, \ldots, C\}} \mathbf{p}_t[c] \tag{6}$$

The detected cells are marked by colored bounding boxes — green and red for normal/typical and infected/abnormal cells, respectively, which not only allows for instant but also easy visual output interpretation.

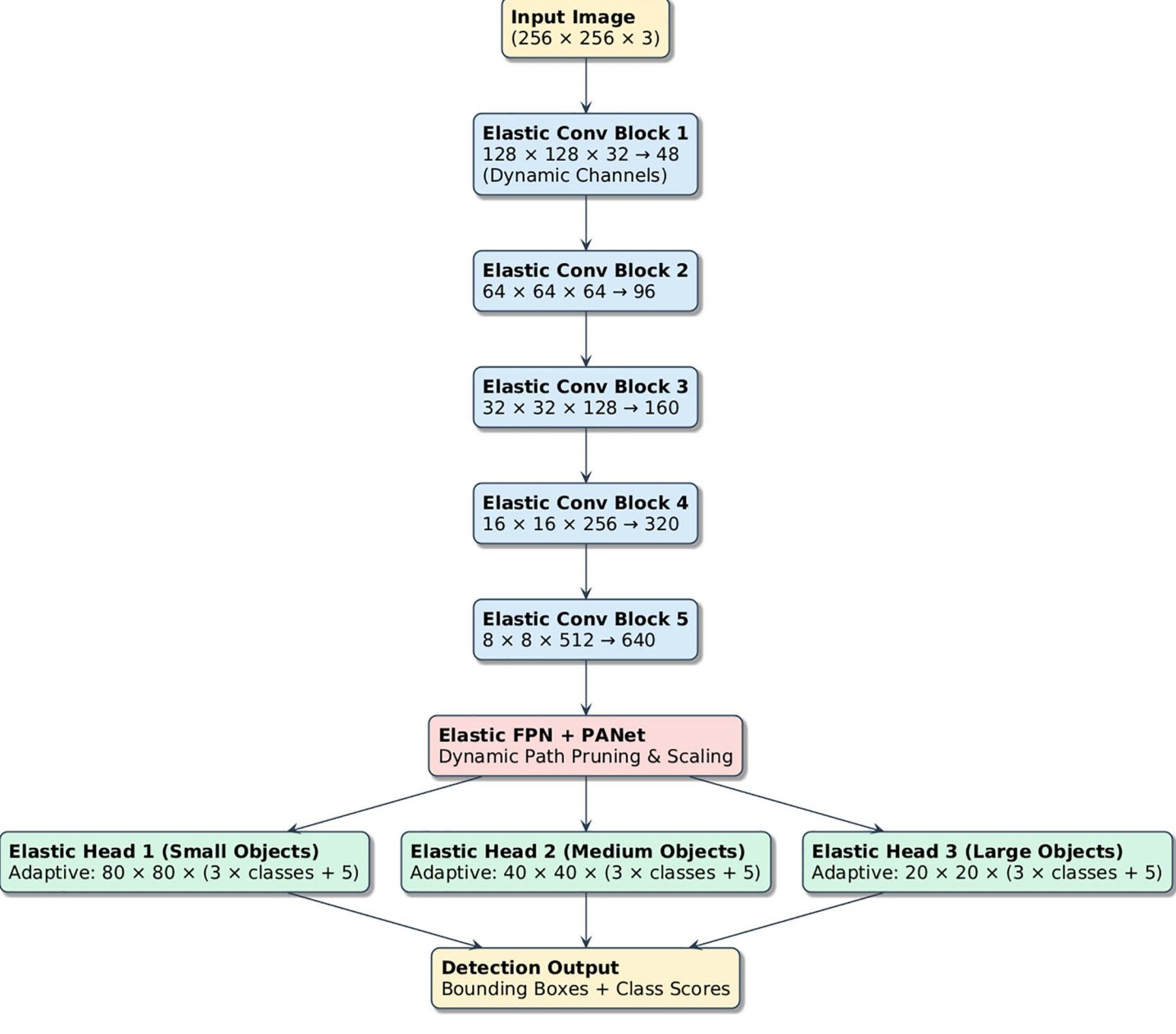

**Fig 4. Proposed Elastic YOLO design integrating adaptive scaling and deformable convolution modules.**

## Performance evaluation Metrics

Metrics used in evaluating the quantitative performance are the F1 Score, Precision, Intersection over Union (IoU), Recall, Mean Average Precision (mAP), Average Precision (AP), and Accuracy (A).

IoU measures the geometric overlap between the predicted bounding box and the ground-truth bounding box. AP represents the area under the precision–recall curve for a given class. mAP is computed as the mean of the AP values across all classes and is commonly used to evaluate the overall performance of object detection systems.

These 7 metrics collectively assess the quantitative performance of the Elastic YOLO system. Together, these metrics provide a comprehensive overview of detection accuracy in terms of both spatial localization (through IoU and precision–recall relationships) and semantic classification reliability across different blood cell categories.

F1-score, Recall, and Precision are computed as in (7):

$$F_1 = \frac{2PR}{P + R}, \quad R = \frac{\mathcal{C}_{\text{pos}}}{\mathcal{C}_{\text{pos}} + \mathcal{M}_{\text{neg}}}, \quad P = \frac{\mathcal{C}_{\text{pos}}}{\mathcal{C}_{\text{pos}} + \mathcal{M}_{\text{pos}}}, \tag{7}$$

$\mathcal{C}_{\text{pos}}$ refers to instances of correctly identified Regions-of-Interest (ROIs), while $\mathcal{M}_{\text{pos}}$ indicates misidentified ROIs (false positives) and $\mathcal{M}_{\text{neg}}$ indicates undetected ROIs (false negatives). Therefore, precision indicates the proportion of accurate ROIs detected as compared to total predictions; recall indicates the proportion of total detections from true ROIs; and $F_1$ provides an average of these two metrics produced by the above method to allow for equitable treatment across categories in cases where they are imbalanced.

The mean average, and average precision values are calculated as in (8):

$$mAP = \frac{1}{C} \sum_{i=1}^{C} AP_i, \quad AP = \int_0^1 P(R) \, dR \tag{8}$$

with $C = 6$, the total number of morphological categories. $AP$ is the area under the curve of P–R for each class, and $mAP$ is the global consistency of detection accuracy for all the cell types combined (RBC, ARBC, WBC, AWBC, PL, APL).

Accuracy and IoU are formulated in (9):

$$A = \frac{\mathcal{C}_{\text{pos}} + \mathcal{C}_{\text{neg}}}{\mathcal{C}_{\text{pos}} + \mathcal{C}_{\text{neg}} + \mathcal{M}_{\text{pos}} + \mathcal{M}_{\text{neg}}}, \quad IoU = \frac{\text{Area of Overlap}}{\text{Area of Union}} \tag{9}$$

where $\mathcal{C}_{\text{neg}}$ denotes correctly ignored background regions. The accuracy is an assessment of the overall correctness of detections across all categories, while IoU measures the geometric overlap between ground-truth & predicted bounding boxes — essential for dense smear regions with overlapping cells.

## Loss Function Formulation

The proposed Elastic YOLO model utilizes a joint loss function for better detection accuracy, category discrimination, and deformation consistency, as given in (10):

$$L_{\text{total}} = L_{\text{reg}} \lambda_{\text{reg}} + L_{\text{cls}} \lambda_{\text{cls}} + L_{\text{det}} \lambda_{\text{det}} \tag{10}$$

here $\lambda_{\text{reg}} = 0.05$, $\lambda_{\text{cls}} = 1.0$, $\lambda_{\text{det}} = 1.0$. localization and class prediction performance are regulated by the first two terms, and the final term suppresses over-deformation in elastic convolution offsets.

For localizing the object, the Complete Intersection over Union (CIoU) loss is given in (11):

$$L_{\text{CIoU}} = 1 - IoU + \alpha v + \frac{\rho^2(b^*, b)}{c^2} \tag{11}$$

where $\rho(b, b^*)$ is the euclidean distance between the centers of the ground-truth $b^*$ and its predicted box $b$, $v$ is the aspect ratio deviation, $c$ is the minimum enclosing box, and $\alpha$ is a balancing factor. This improves the accuracy of localization and shape consistency of the boxes for cell boundaries of complex or irregular shapes.

As given in (12), the categorical loss follows a cross-entropy structure:

$$L_{cls} = -\sum_{i=1}^{C} \log(\hat{y}_i) y_i$$

(12)

Here, $\hat{y}_i$ and $y_i$ refer to the predicted and actual probabilities of the $i^{th}$ class. This ensures proper mapping to the corresponding class of objects.

The elastic regularization constraint is given by (13):

$$L_{reg} = \|\Delta p\|_2^2$$

(13)

This constrains the offset vectors $\Delta p$ for the deformable convolution kernels, thereby stabilizing spatial adaptability without compromising structural consistency.

## Model complexity analysis

Floating-point operations (FLOPs) are used to estimate the computational cost of the neural network. As illustrated in (14), the computational complexity of the Elastic YOLO model is quantified in terms of FLOPs:

$$FLOPs = \sum_{l=1}^{L} 2\, C_{in}^l C_{out}^l K_h^l K_w^l H_{out}^l W_{out}^l$$

(14)

This is defined as the sum of the channel numbers $C_{in}^l$ and $C_{out}^l$ for each layer in the network. This is done for the total number of layers in the network, i.e., $L$. This includes the kernel sizes $K_w^l$ and $K_h^l$ for each layer. It also includes the output sizes $W_{out}^l$, $H_{out}^l$.

For an input size of $256 \times 256$ pixels, the network achieves 78 frames per second (FPS) with a computational cost of 30.1 GFLOPs and 12.4M parameters. This is achieved using an NVIDIA 4090 GPU. This requires roughly 110 MB of memory. This is a little larger than the YOLOv5 network. However, the dynamic pruning and adaptive WD scaling ensure real-time performance with the morphological complexities present in the dense cell images.

## Algorithmic representation and complexity evaluation

The Elastic YOLO processing flow is described in Algorithm 3. The algorithm highlights the key phases: preprocessing and augmentation, feature extraction in an adaptive manner using elastic convolution, hierarchical feature fusion, and detection and visualization that is morph-aware (Table 3).

## Computational complexity analysis

Let $C_{in}^{(l)}$ and $C_{out}^{(l)}$ be the input and output channels of the layer $l$, $L$ be the total number of convolutional operations, and $K$ the kernel size. The impact of pruning and dynamic scaling operations has an impact on the complexity performance.

**Time Complexity:**

$$\mathcal{T}(N) = O\left(\sum_{l=1}^{L} C_{in}^{(l)} \times C_{out}^{(l)} \times K^2 \times H_{out}^{(l)} \times W_{out}^{(l)}\right)$$

**Table 3. The Workflow of the Elastic YOLO Model for Blood Cell Morphology Detection.**

| Step | Description |
|------|-------------|
| **Inputs and Parameters** | |
| – | Number of classes $N_c$ = 6, IoU threshold $\tau_{IoU}$ = 0.5, and input image $\mathbf{X} \in \mathbb{R}^{256 \times 256 \times 3}$. |
| **Stage 1: Preprocessing** | |
| 1 | Channel normalization: $\mathbf{X}' = (\mathbf{X} - \bar{\mu})/\bar{\sigma}$. |
| 2 | Apply random geometric transformations: scaling $\varsigma \sim U[0.9, 1.1]$, translation $(t_x, t_y) \sim U[-0.1R, 0.1R]$, and rotation $\phi \sim U[-15°, 15°]$ to obtain transformed sample $\mathbf{X}''$. |
| **Stage 2: Elastic Feature Extraction (Backbone)** | |
| 3 | Initialize elastic scaling coefficients $\lambda_d$ (depth) and $\lambda_w$ (width). |
| 4 | For each backbone stage $s \in \{1, 2, 3, 4, 5\}$ and each layer $l = 1$ to $\lambda_d \cdot D_s$: $\mathbf{F}_{s,l} = \text{SiLU}\big(\text{BN}(\text{EConv}(\mathbf{F}_{s,l-1}; \lambda_w, \Delta\mathbf{p}))\big)$ where EConv denotes elastic convolution with spatial offsets $\Delta\mathbf{p}$. |
| 5 | Store multi-scale feature outputs $\{\mathbf{F}_s^{(1)}, \mathbf{F}_s^{(2)}, \ldots\}$ for fusion. |
| **Stage 3: Hierarchical Fusion (Elastic FPN + PANet)** | |
| 6 | For each fusion level $k$, compute: $\mathbf{U}_k = \text{SiLU}\big(\text{BN}(\text{EConv}(\text{Fuse}(\mathbf{F}_{k-1}, \mathbf{F}_k)))\big)$ and remove paths contributing less than threshold $\epsilon$ via adaptive pruning. |
| **Stage 4: Detection Head (Anchor-Free Prediction)** | |
| 7 | For each feature scale $r$, predict class logits $\mathbf{Z}_r$ and regression offsets $\mathbf{R}_r$. |
| 8 | Decode regression outputs to bounding boxes $b_i = (\tilde{x}_i, \tilde{y}_i, \tilde{w}_i, \tilde{h}_i)$ and compute class probabilities $\tilde{p}_i = \text{Softmax}(\mathbf{Z}_r)$. |
| 9 | Aggregate all predictions: $\mathcal{B} = \bigcup_r \{(b_i, \tilde{p}_i)\}$. |
| **Stage 5: Post-processing and Visualization** | |
| 10 | Apply Non-Maximum Suppression (NMS) such that a box $b_i$ is retained if $\text{IoU}(b_j, b_i) < \tau_{IoU}$ for all higher-confidence boxes $b_j$. |
| 11 | Render bounding boxes: abnormal classes {AWBC, APL, ARBC} in red, and normal classes in green. |
| **Output** | |
| – | Final image annotated with bounding boxes $\mathcal{B}$. |

Elastic pruning is included within the fusion block to minimise redundancy by reducing the number of times calculations need to be performed, therefore reducing the number of FLOPs relative to YOLOv5.

**Space Complexity:**

$$\mathcal{S}(N) = O\left(\sum_{l=1}^{L} H_{\text{out}}^{(l)} \times W_{\text{out}}^{(l)} \times C_{\text{out}}^{(l)}\right) + O(\Theta)$$

Where $\Theta$ is the total number of trainable parameters, including elastic scaling coefficients and deformable offset values.

**Model Profile and Efficiency Metrics:**

• Throughput: **78 FPS** (on NVIDIA RTX 4090)

• FLOPs: **30.1G** for $256 \times 256$ input resolution

• Model footprint: $\approx$ 110 **MB**

• Parameter count: **12.4M**

Incorporating both elastic convolutional scaling and adaptive pruning provides optimal computational efficiency without limiting the robustness of the detection system, thereby providing an improved balance of accuracy and real-time inference capability.

## Results and discussion

This section presents the experimental results of the Elastic YOLO approach proposed to classify and detect abnormal and normal white, red blood cells, platelets, their morphological invariants, and their variants in images obtained from peripheral blood smears. They are based on quantitative measurement, qualitative examples, comparative evaluation, and ablation studies.

### System configuration

The test was performed on a PC with 64 GB RAM, an Intel Core i9-13900K CPU, CUDA 12, NVIDIA RTX 4090 GPU (24 GB VRAM), Python 3.10, Ubuntu 22.04, and PyTorch 2.1. The annotation dataset was divided into 15% testing, 15% validation, & 70% train. It was trained for 100 epochs with Elastic YOLO and the parameters defined in Section Proposed Methodology.

### Baseline vs. Proposed elastic YOLO

The detection accuracy per-class over normal/regular and abnormal WBC, PL, and RBC on the proposed Elastic YOLO and the baseline YOLOv5 is tabulated in Table 4. The proposed scheme shows consistent gains over the selected baseline on all the classes, heavily on the abnormal cells.

The abnormal categories (ARBC, AWBC, and APL) demonstrate consistent improvements in detection performance compared with the baseline YOLOv5 model. This is particularly important in clinical hematology workflows, where abnormal cells occur less frequently but carry higher diagnostic significance.

The Fig 5 presents the detection outputs. YOLOv5 occasionally misses the overlapping or shape-transformed cells, while the Elastic YOLO recognizes and labels them with better accuracy. It correctly matches the ground truth in 93.4% of test images and occasionally is incorrect in small cells, displaying increased robustness.

**Table 4. Per-class detection metrics for Elastic YOLO and Baseline YOLOv5.**

| Class | Model | R (%) | mAP@0.5 (%) | P (%) |
|---|---|---|---|---|
| RBC (Normal) | Elastic YOLO | 94.7 | 95.5 | 96.1 |
| | Baseline YOLOv5 | 92.8 | 93.5 | 94.2 |
| RBC (Abnormal) | Elastic YOLO | 91.7 | 92.8 | 93.9 |
| | Baseline YOLOv5 | 88.5 | 89.2 | 90.4 |
| WBC (Normal) | Elastic YOLO | 95.2 | 96.1 | 97.0 |
| | Baseline YOLOv5 | 93.6 | 94.4 | 95.5 |
| WBC (Abnormal) | Elastic YOLO | 92.3 | 93.4 | 94.5 |
| | Baseline YOLOv5 | 89.1 | 90.3 | 91.8 |
| PL (Normal) | Elastic YOLO | 93.2 | 94.1 | 95.0 |
| | Baseline YOLOv5 | 91.4 | 92.2 | 93.1 |
| PL (Abnormal) | Elastic YOLO | 90.1 | 91.3 | 92.4 |
| | Baseline YOLOv5 | 87.0 | 88.3 | 89.7 |

R, P, and mAP@0.5, define Recall (R), Precision (P), and Mean Average Precision (mAP), respectively, at an Intersection over Union (IoU) threshold of 0.5.

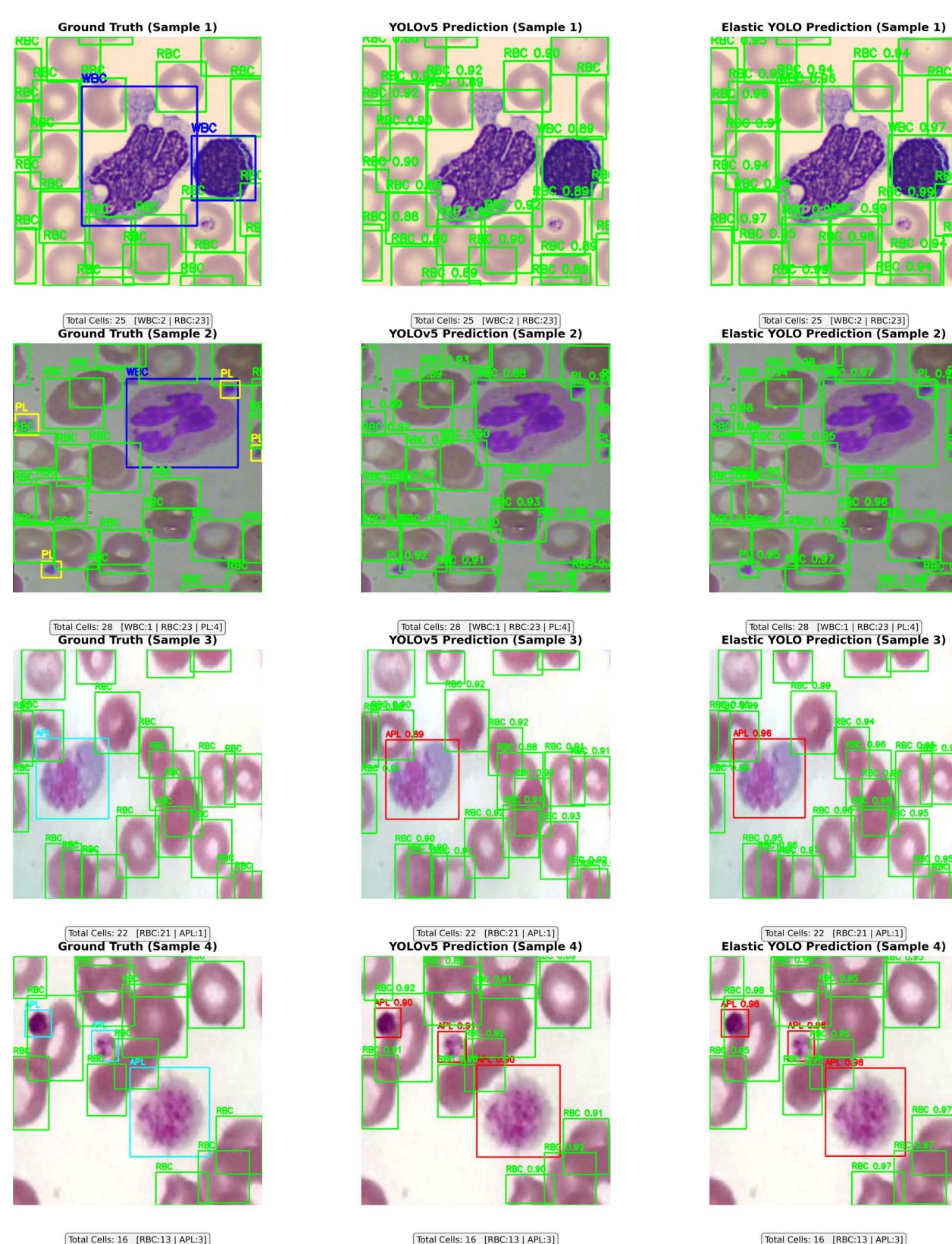

**Fig 5. Detection results on test samples for YOLOv5 and Elastic YOLO.**

One row represents one individual image, showing the Ground Truth along with the elastic YOLO Predictions and YOLOv5 Predictions, with class-specific bounding boxes—using red for abnormal-type cells (AWBC, APL, ARBC), green for normal-type cells (WBC, PL, RBC).

Alongside each recognition are the referring confidence scores. At the bottom of each image are numerical summaries that detail the detection rates per class and the overall counts. The elastic YOLO presents high confidence and increased robustness in detecting subtle morphologic irregularities, like large platelets, atypical white-blood cells, anisocytosis in red-blood cells, and correctly identifying normal cells.

### IoU evaluation

Table 5 compares IoU on every class. Elastic YOLO's results are always better, particularly on irregular-shaped abnormal RBCs and WBCs, whose localization is challenging due to the irregularity of their shape and the artifact of staining.

### Average Precision (AP) Across multiple IoU thresholds

Table 6 contrasts AP at COCO mAP@0.5:0.95 IoU, 0.75, and 0.5 thresholds. AP is greater in the Elastic YOLO on every one of these conditions, as can be seen in the Fig 6 below.

### Analysis of loss trends

The validation and training loss curves are shown in Fig 7. Where the elastic YOLO converges sooner and has lower focus loss (better control of class imbalance) and losses due to elastic regularization (better control of deformations).

### Performance across training epochs

The accuracy and the loss values at epochs 100, 75, 50, 25 are tabulated in Tables 7 and 8, and the respective loss convergence is depicted in Fig 7.

**Table 5. IoU comparison between Elastic YOLO and Baseline YOLOv5.**

| Class | Elastic YOLO | Baseline YOLOv5 |
|---|---|---|
| WBC (Normal) | 91.8 | 89.5 |
| WBC (Abnormal) | 88.2 | 85.0 |
| RBC (Normal) | 90.4 | 88.1 |
| RBC (Abnormal) | 93.9 | 84.2 |
| PL (Normal) | 89.6 | 87.3 |
| PL (Abnormal) | 86.0 | 83.1 |

The values represent an average of Intersection over Union (IoU) scores (%) that were obtained across the test dataset for normal and abnormal blood cells.

**Table 6. AP comparison across multiple IoU thresholds.**

| Model | mAP@0.5:0.95 (%) | AP@0.75 (%) | AP@0.5 (%) |
|---|---|---|---|
| Elastic YOLO | 87.8 | 89.6 | 94.7 |
| Baseline YOLOv5 | 85.0 | 87.1 | 92.3 |

AP stands for Average Precision, and has been assessed at various Intersection over Union (IoU) thresholds following the Common Object Detection Results (COCO) evaluation protocol.

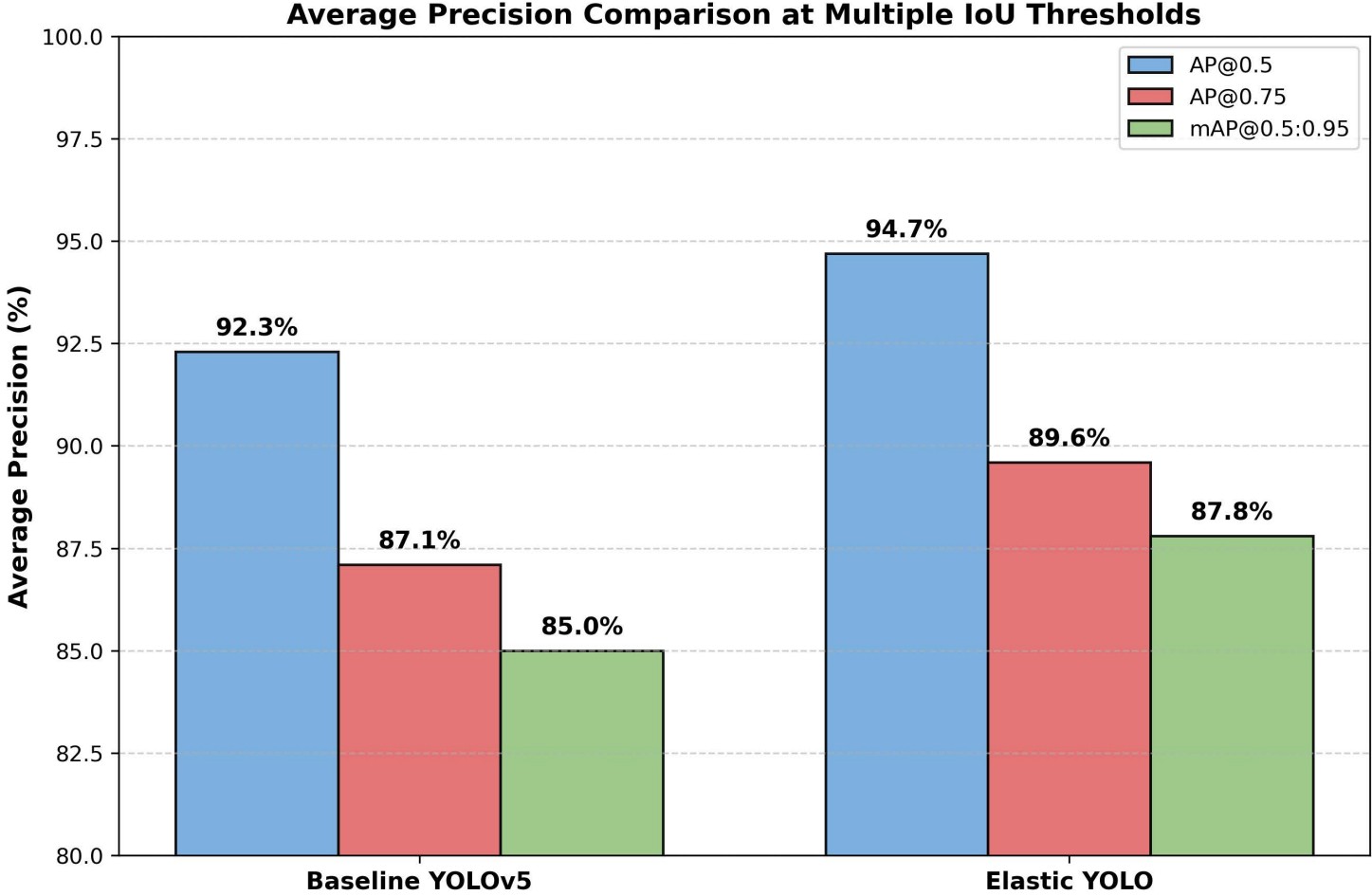

**Fig 6. AP comparison at multiple IoU thresholds.**

Elastic YOLO showed consistent performance gains relative to the baseline YOLOv5 in terms of accuracy on the training and testing sets (up to 96.0% and 94.0%, respectively) as well as a reduced total loss ($L_{total}$), as well as low individual loss terms ($L_{reg}$, $L_{cls}$, $L_{det}$,) at each checkpoint.

Table 8 illustrates that the EYOLO model exhibits a smoother and more pronounced decline in loss (0.74$\rightarrow$0.56) compared with YOLOv5 (0.84$\rightarrow$0.69) over 25–100 epochs.

The ongoing enhancement, as demonstrated by the loss curves illustrated in Fig 7, indicates a more effective convergence in optimization.

As the error between training and test accuracy is always small throughout the process of training, there are no signs of overfitting.

**Comparison with SOTA Methods**

Table 9 shows that, although YOLOv8 produces the maximum FPS (85), EYOLO achieves the best mAP@0.5 (94.7%), recall (93.6%), and precision (95.1%), and achieved a 1.3% increase in recall compared with YOLOv8, and 1.1% mAP. FLOPs are 30.1 G, and the model size is 110 MB.

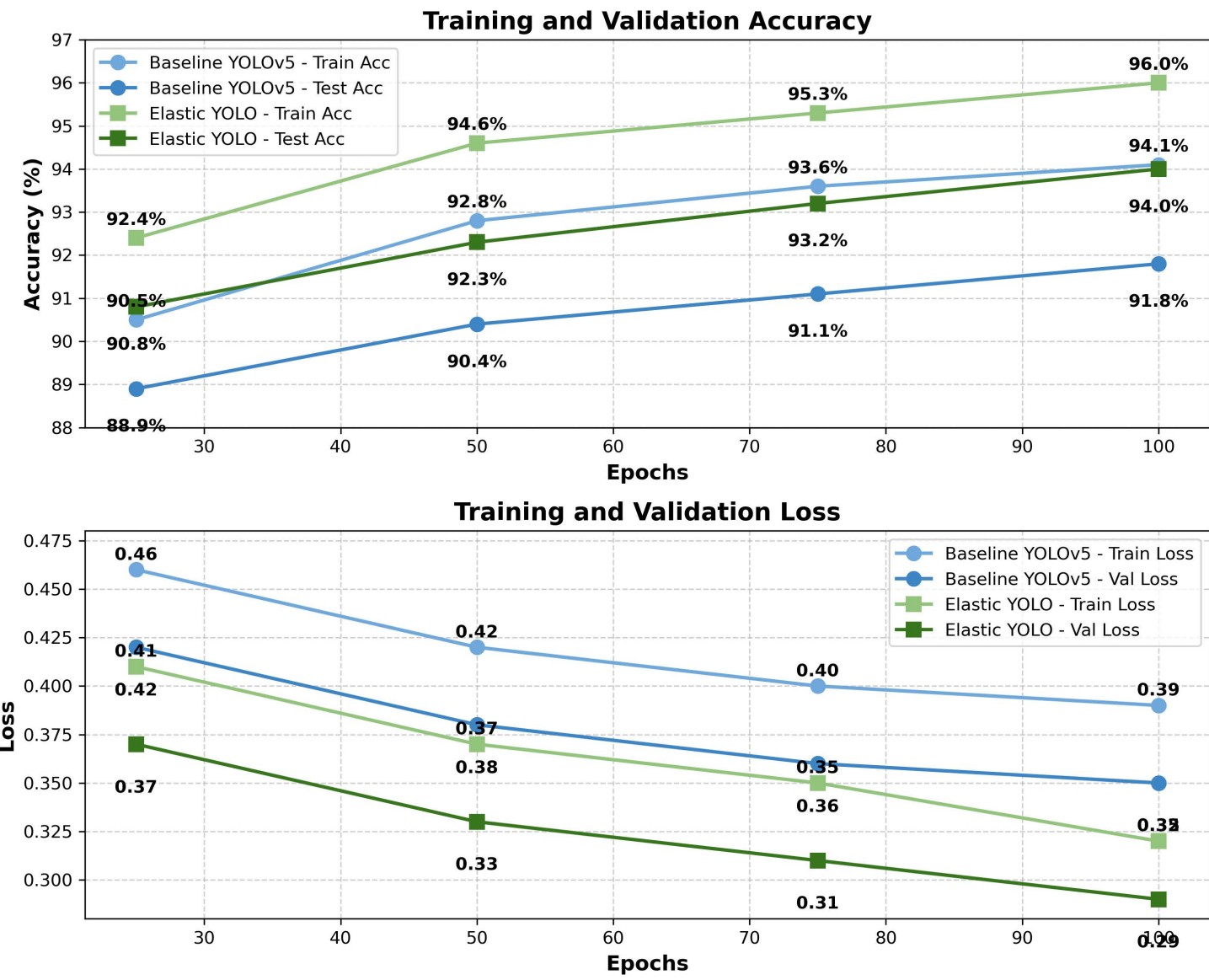

**Fig 7. Training and validation accuracy-loss trends across epochs for EYOLO and YOLOv5 models.**

To substantiate these findings, Fig 8 illustrates qualitative comparisons regarding the detection performance of Elastic YOLO among various SOTA models, such as YOLOv3–v8, RetinaNet, and Faster RCNN. The Elastic YOLO Model has been found to produce the highest level of bounding box location accuracy and highest confidence score (correctly state abnormal/presumed) for the separation of abnormal and presumed platelet types (APL) when compared to abnormal/presumed WBCs (AWBC). The results presented in Table 9 further validate the quantitative advantages of using the Elastic YOLO Model.

In addition to the qualitative metrics shown above, the Fig 9 also provides a graphical comparison of (a) the quantitative performance results and (b) speed of computation, demonstrating the ability of the Elastic YOLO Model to achieve a balance between accuracy and speed when used for blood cell image morphometry analysis.

**Table 7. Epoch-wise comparison of training and validation metrics for Elastic YOLO and YOLOv5.**

| Epoch | Model | Train Loss | Train Acc. (%) | Val. Loss | Test Acc. (%) |
|---|---|---|---|---|---|
| 25 | EYOLO | 0.41 | 92.4 | 0.37 | 90.8 |
|  | YOLOv5 | 0.46 | 90.5 | 0.42 | 88.9 |
| 50 | EYOLO | 0.37 | 94.6 | 0.33 | 92.3 |
|  | YOLOv5 | 0.42 | 92.8 | 0.38 | 90.4 |
| 75 | EYOLO | 0.35 | 95.3 | 0.31 | 93.2 |
|  | YOLOv5 | 0.40 | 93.6 | 0.36 | 91.1 |
| 100 | EYOLO | 0.32 | 96.0 | 0.29 | 94.0 |
|  | YOLOv5 | 0.39 | 94.1 | 0.35 | 91.8 |

Train and validation loss values indicate convergence behavior, while test accuracy reflects generalization performance across different training epochs.

**Table 8. Epoch-wise analysis of individual and total loss for Elastic YOLO and YOLOv5.**

| Epoch | Model | $L_{det}$ | $L_{cls}$ | $L_{reg}$ | $L_{total}$ |
|---|---|---|---|---|---|
| 25 | EYOLO | 0.16 | 0.21 | 0.37 | 0.74 |
|  | YOLOv5 | 0.18 | 0.24 | 0.42 | 0.84 |
| 50 | EYOLO | 0.14 | 0.19 | 0.33 | 0.66 |
|  | YOLOv5 | 0.16 | 0.22 | 0.38 | 0.76 |
| 75 | EYOLO | 0.13 | 0.17 | 0.31 | 0.61 |
|  | YOLOv5 | 0.15 | 0.21 | 0.36 | 0.72 |
| 100 | EYOLO | 0.12 | 0.15 | 0.29 | 0.56 |
|  | YOLOv5 | 0.14 | 0.20 | 0.35 | 0.69 |

$L_{det}$, $L_{cls}$, and $L_{reg}$ define the detection, classification, and regression loss components of the model respectively, whereas $L_{total}$ means the weighted summation of all three components.

**Table 9. Benchmarking Elastic YOLO against SOTA models on the dataset.**

| Model | mAP@0.5 (%) | P (%) | R (%) | FPS | Params (M) | FLOPs (G) | Size (MB) |
|---|---|---|---|---|---|---|---|
| Elastic YOLO | 94.7 | 95.1 | 93.6 | 78 | 12.4 | 30.1 | 110 |
| YOLOv8 | 93.6 | 94.2 | 92.3 | 85 | 11.1 | 28.3 | 95 |
| YOLOv7 | 93.1 | 94.0 | 91.8 | 70 | 36.9 | 104.7 | 72 |
| YOLOv5 | 91.4 | 93.0 | 90.1 | 62 | 7.2 | 16.5 | 98 |
| YOLOv4 | 90.7 | 92.4 | 89.6 | 58 | 64.4 | 164.8 | 244 |
| YOLOv3 | 89.3 | 91.2 | 88.1 | 45 | 61.9 | 155.2 | 235 |
| RetinaNet | 88.6 | 90.3 | 87.5 | 35 | 36.5 | 179.3 | 145 |
| Faster R-CNN | 86.8 | 88.5 | 85.7 | 25 | 42.1 | 207.4 | 240 |

Params are the number of model parameters in millions, and FLOPs indicate the amount of computation required to execute the model. Size means the amount of space occupied by a model on disk.

## Ablation analysis

The function of each individual component of the Elastic YOLO model is summarised by means of the above mentioned Fig 10 as well as Table 10.

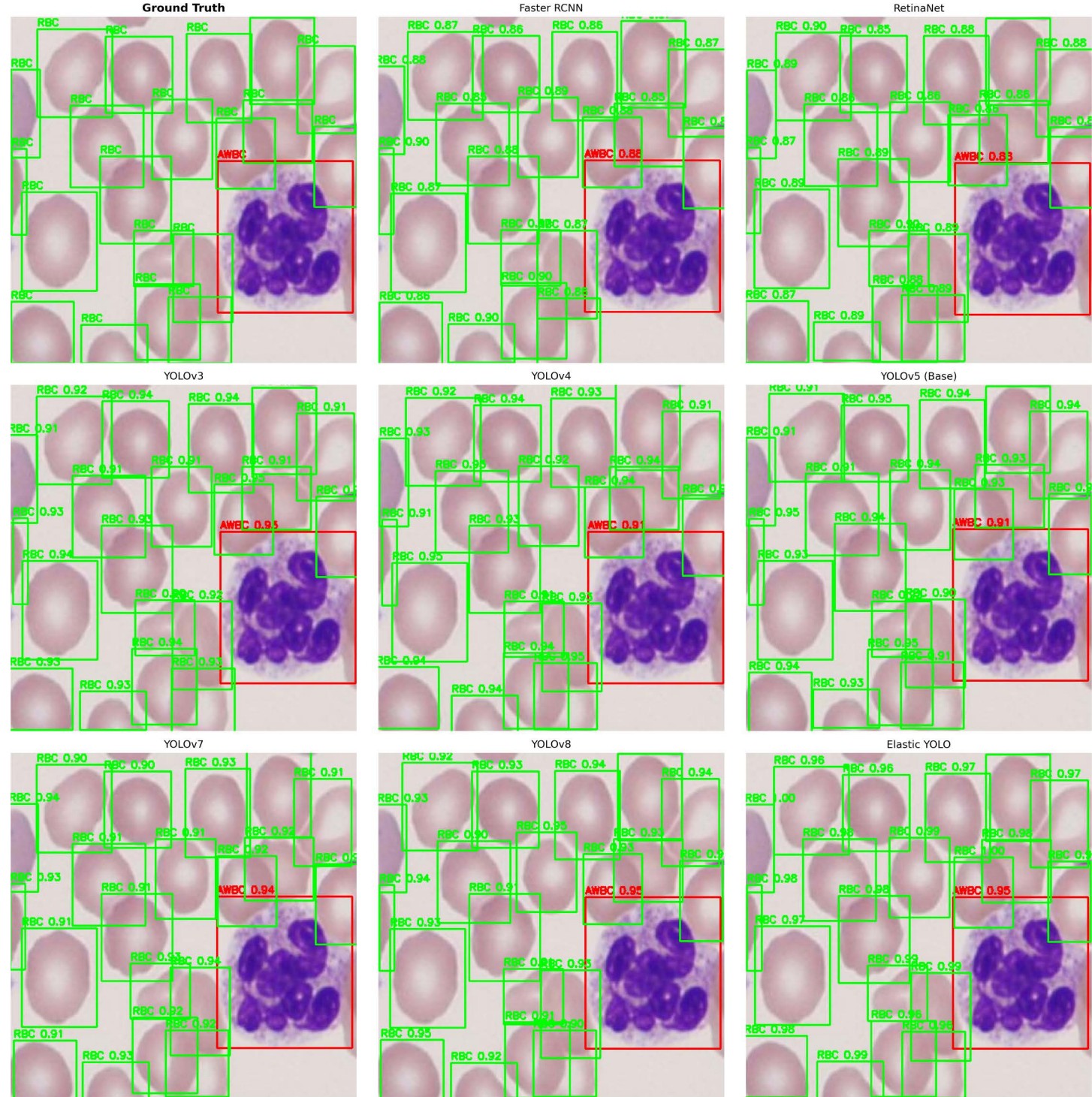

**Fig 8. Qualitative detection comparison of various models on a sample microscopy image.** Red: abnormal cells; green: normal cells.

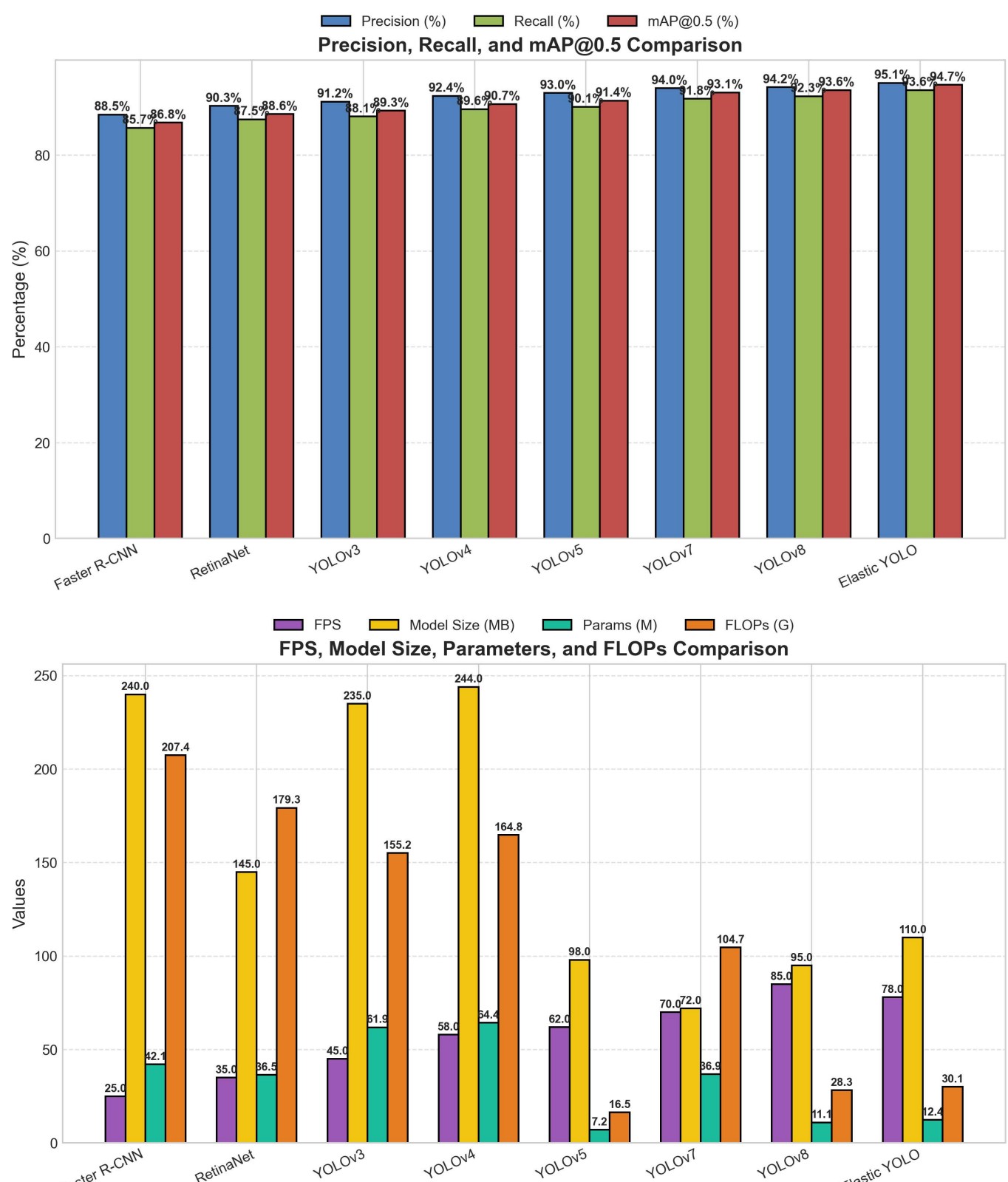

**Fig 9. State-of-the-art model comparison. (A)** Evaluation metrics. **(B)** Computational complexity.

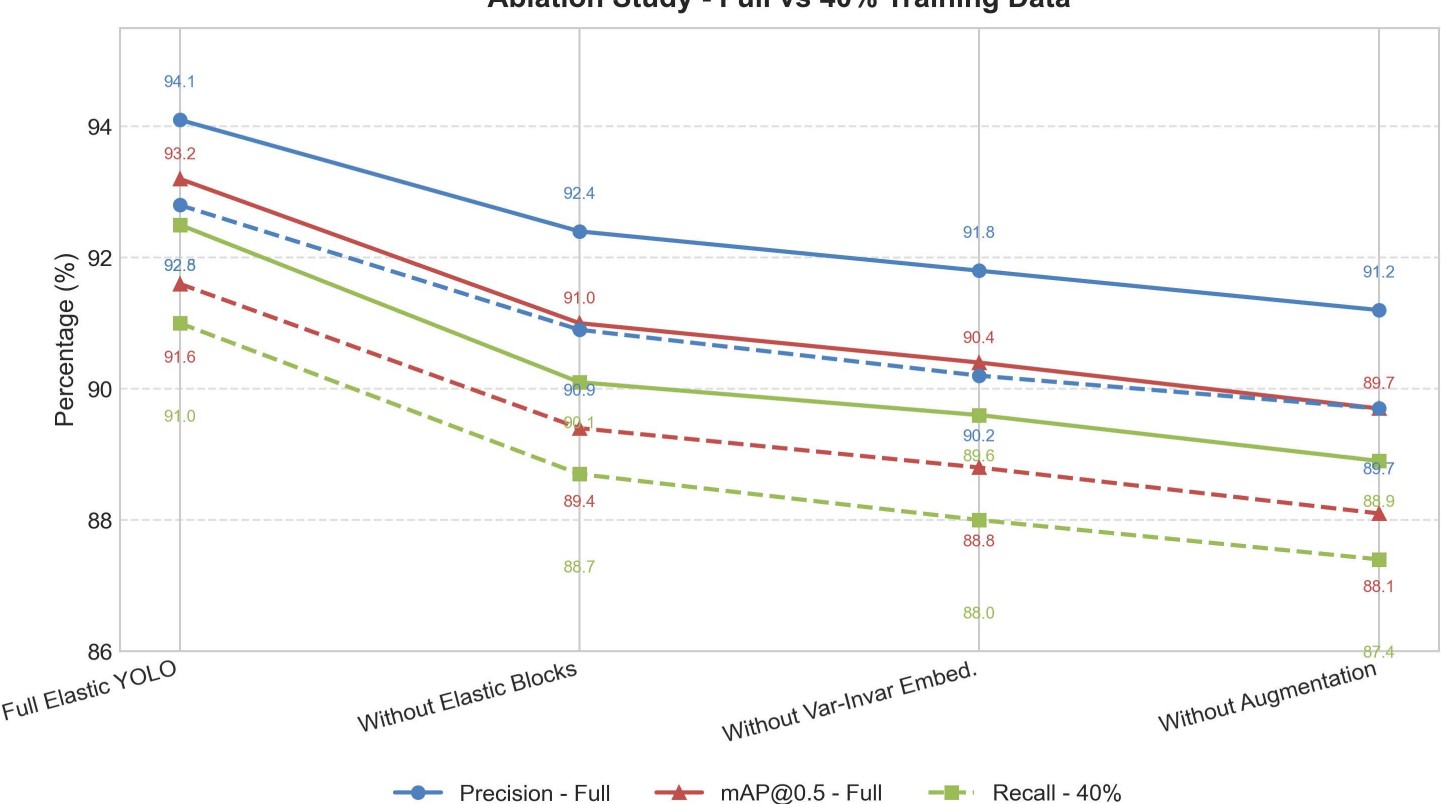

**Fig 10. Ablation analysis results.**

**Table 10. Ablation study results for Elastic YOLO.**

| Configuration | mAP@0.5 (%) | R (%) | P (%) |
|---|---|---|---|
| Without augmentation | 91.0 | 89.8 | 92.2 |
| Without Elastic blocks | 92.3 | 91.2 | 93.4 |
| Without Var-Invar Embed. | 91.7 | 90.5 | 92.7 |
| Full Elastic YOLO | 94.7 | 93.6 | 95.1 |

The basis of the ablation study is to evaluate the contribution of data augmentation and elastic convolution blocks, as well as confidential invariant embedding, to improve overall detection performance.

The omission of variant-invariant embedding, or elastic convolution, blocks will negatively impact performance accuracy and confidence with respect to abnormal cells. Also, in the absence of data augmentation, the robustness to variations in lighting and staining decreases.

In addition to component ablation, sensitivity analysis was conducted for the elastic scaling parameters governing depth ($\lambda_d$) and width ($\lambda_w$). Moderate variations within the predefined ranges ($\lambda_d \in \{0.75, 1.0, 1.25\}$ and $\lambda_w \in \{0.5, 1.0, 1.5\}$) resulted in stable detection performance, with mAP fluctuations remaining within ±1.2%. This indicates that the proposed elastic design is robust to parameter selection and does not rely on narrowly tuned architectural configurations.

From the data shown in the Table 11, even with only 40% of a training dataset, the Elastic YOLO Model is still able to demonstrate improved performance over its ablated variations, even under conditions of reduced dataset availability, which is a key indicator of reliable peripheral blood smear imaging performance.

## Qualitative results

Fig 5 depicts a comparative evaluation between detection results for four picked peripheral blood smear images tested under varying microscopy conditions.

The interface for detecting prototypes is given as Fig 11a and is composed of bounding boxes predicted, an auto-generated analysis summary (abnormal/normal distribution, cell counts, dominant abnormality, and suggestion towards diagnosis), along with the input images.

The exported diagnostic report shown as Fig 11b consolidates annotated detections with statistical summaries that can be put into documentation or read off-site.

## Error analysis

Failure cases in Fig 12, include missed detections, similar neighboring clusters, low contrast, and overlapping cells.

On occasion, small platelets and abnormal RBCs are not detected. Key factors are the quality of the image and the dataset. This phenomenon is further influenced by the general class imbalance of peripheral blood smear images, where RBCs are prominent, and abnormal cells of clinical interest are rare.

Proposed enhancements include the integration of attention mechanisms to deal with occlusions and the use of higher-magnification images to improve edge detection.

Elastic YOLO demonstrates higher detection reliability than the baseline YOLOv5 and other evaluated models.

These failure cases primarily reflect inherent challenges in microscopic imaging rather than instability in the proposed framework.

## Confusion matrix analysis

To further evaluate classification reliability across the six blood cell categories, a confusion matrix was constructed using predictions from the test dataset. The matrix summarizes the agreement between ground-truth labels and predicted classes for RBC, WBC, platelets, and their abnormal variants.

The confusion matrix presented in Fig 13 reveals strong diagonal dominance, indicating accurate class predictions. Most misclassifications occur between morphologically similar categories, particularly between normal and abnormal red blood cells (RBC and ARBC) and between platelet variants (PL and APL). These errors are primarily attributed to overlapping cell boundaries and subtle staining variations, which remain challenging for automated detection systems.

Overall, the confusion matrix confirms the robustness of the proposed Elastic YOLO model for multi-class blood cell detection while highlighting specific morphological categories that require further improvement.

**Table 11. Performance of Elastic YOLO variants with limited training samples.**

| Configuration | mAP@0.5 (%) | R (%) | P (%) |
| --- | --- | --- | --- |
| Without augmentation | 88.1 | 87.4 | 89.7 |
| Without Elastic blocks | 89.4 | 88.7 | 90.9 |
| Without Var-Invar Embed. | 88.8 | 88.0 | 90.2 |
| Full Elastic YOLO | 91.6 | 91.0 | 92.8 |

Based on the results of the ablation study, it is shown that Elastic YOLO remains robust even with small training data, and also has superior generalization capabilities compared to smaller or ablated versions of the same model.

**Fig 11. Elastic YOLO (A) Interface. (B)** Generated diagnostic report.

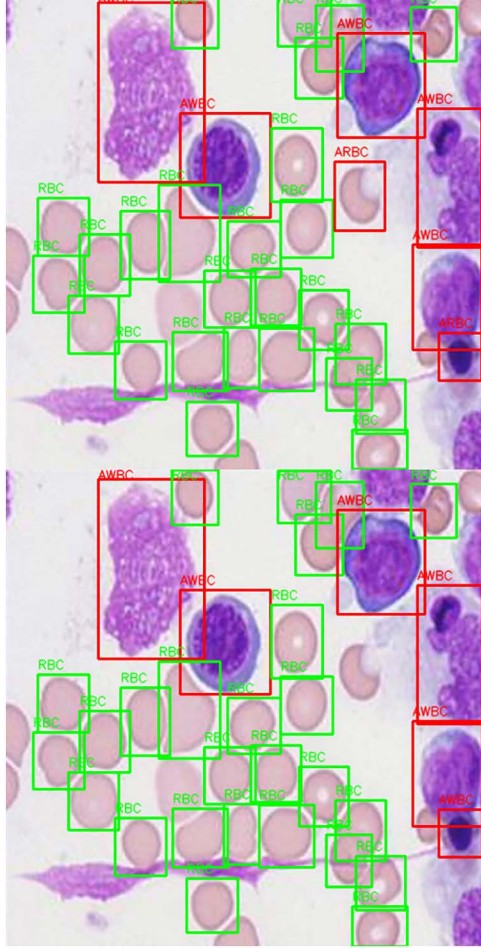

**Fig 12. Failure cases with (A) Ground truth annotations. (B)** Elastic YOLO predictions.

## Discussions

This section combines quantitative results with qualitative and ablation results to situate the Elastic YOLO in the clinical domain. While the model has been trained on datasets, it has been designed to work with other datasets and can be adapted for other peripheral blood smear datasets with minimal retraining.

### Clinical decision support context

The goal of the proposed method, named Elastic YOLO, is to function as a computer-assisted decision support tool. In the context of standard laboratory workflows, automated CBC analysis instruments may prompt peripheral blood smear examinations for flagged abnormalities such as suspected blasts, atypical lymphocytes, platelet abnormalities, or unknown causes of cytopenias. In such situations, the proposed method may assist in identifying and categorizing normal and abnormal types of blood cells on peripheral smear images. This would thus align the proposed framework with real-world laboratory hematology workflows.

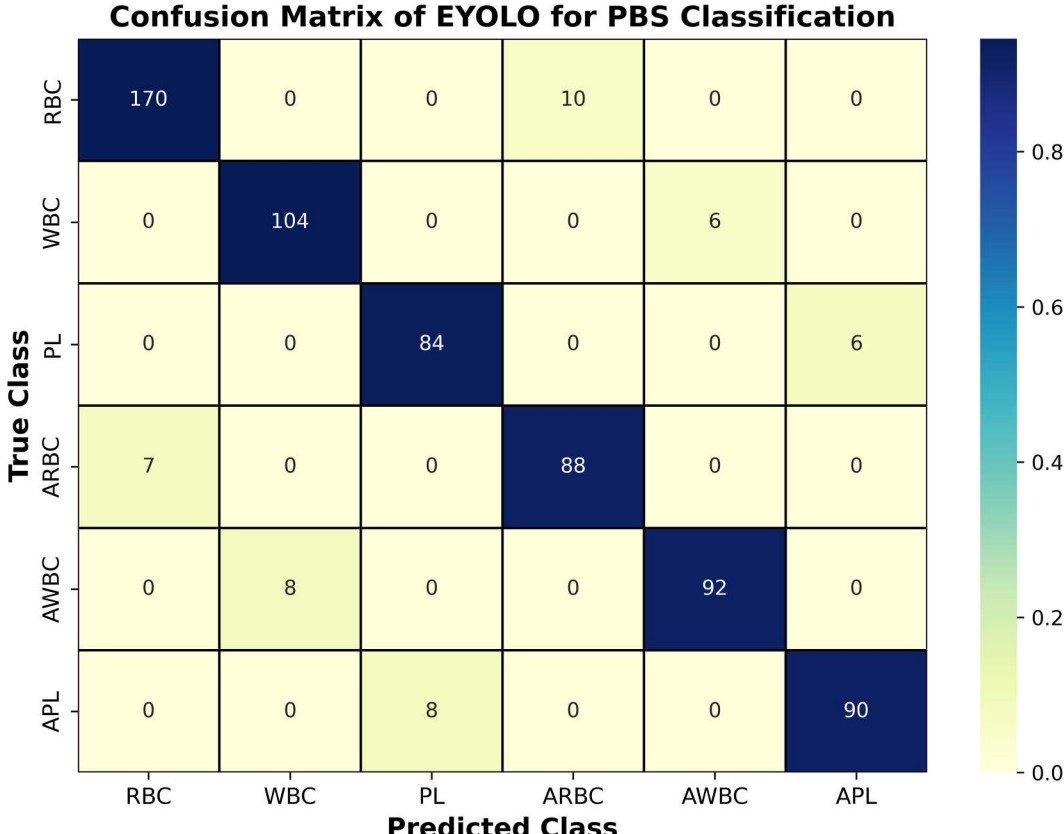

**Fig 13. Confusion matrix illustrating classification performance of Elastic YOLO across six blood cell categories.**

## Clinical challenges beyond routine automated detection

Modern automated hematology analyzers are quite reliable for normal cell types but are still unable to accurately identify some of the clinically important abnormalities, such as leukemic blasts, nucleated red blood cells, basophils, monocytes, and atypical lymphocytes. These cells are either rare, have significant morphologic variability, or are visually difficult to distinguish from other cell types. However, the improved identification of abnormal cell types would suggest that the proposed method, named Elastic YOLO, would perform better for visually identifying abnormalities. Note that subclassification of basophils, monocytes, NRBCs, or blast percentage is not indicated in the study. Future work will focus on extending the dataset and annotation granularity to explicitly assess performance on these diagnostically challenging populations and to quantify potential gains over existing analyzer-based flagging strategies.

## Unified Multi-class morphology detection

Elastic YOLO enables single-shot RBC, WBC, and platelets (PL) detection and classifies normal and abnormal variants in one pass. It addresses the nonavailability of a universal handling workflow dealing with heterogeneous blood cells. Outputs in Tables 4, 7 and Fig 5 show increased accuracy, mAP, and recall, while Fig 11 demonstrates unified detection and reporting.

By jointly detecting RBCs, WBCs, and platelets along with their abnormal variants within a single unified framework, Elastic YOLO directly addresses the limitation of prior methods that focus on isolated cell types or narrowly defined morphological features.

### Detection of morphological variants and invariants

Elastic convolutional blocks with morphology-invariant embeddings increase the capture of subtle and irregular forms such as giant platelets and anisocytosis. Fig 5 and Tables 5–6 confirm the AP and IoU increments between the variants and invariants.

### Dynamic adaptability across architectures

Elastic depth, width, and resolution scaling allow our model to accommodate various devices. As demonstrated by ablation results in Table 10, removing elastic blocks degrades accuracy. In addition, our model's stability under limited data conditions (Table 11) ensures applicability to various computing and medical environments.

### Computational efficiency for Real-time deployment

As demonstrated in Table 9, our model achieves a satisfactory balance between speed and model size: 78 FPS and 110 MB. This makes Elastic YOLO applicable to real-time analyzers and telemedicine platforms.

Elastic YOLO was primarily designed to support lightweight model deployment. As demonstrated in Table 10, even after applying elastic convolution and scaling, our model's 12.4M parameters and 30.1 GFLOPs are efficient. In addition, it achieves 78 FPS on an NVIDIA RTX 4090 GPU. These results indicate our model's potential to support real-time analysis and applications in hematology.

### Data efficiency and generalization

Augmentation and transfer learning reduce the necessity to use large datasets. As demonstrated in Table 11, our model's mAP loss is minimal even when using 60% fewer samples. These results indicate our model's robustness and applicability to various image sources and stains.

### Error analysis and future directions

Currently, the proposed model's limitations are due to soft cell boundaries, overlapping cell groups, and poor image contrast. These factors lead to missed detections, particularly small platelets and faint abnormal RBCs (Fig 12). Future directions will involve using Explainable AI (XAI), high-resolution processing, and attention modules to improve model accuracy and increase clinician confidence.

## Conclusion

The Elastic YOLO model represents a powerful and clinically relevant solution for localizing, detecting, and classifying peripheral blood smear cells, both abnormal and normal/typical. The Elastic YOLO model utilizes the power of the YOLOv5 framework to incorporate the elastic convolutional blocks to ensure the model can cope with fine-grained morphological differences due to diverse imaging conditions. The experimental results show the superiority of the Elastic YOLO model in terms of improved performance compared to the baseline YOLOv5 and state-of-the-art models in terms of mAP, recall, precision, and IoU. At the same time, the Elastic YOLO model retains the advantage of a good balance in terms of detection accuracy, computational speed, and model size.

This system may be useful in helping clinicians analyze blood samples in which the blood cells may be of varying shapes. This is a situation that may be commonly encountered in routine clinical practice. The system does not require any demographic details of the patient who is to be examined.

Thus, the Elastic YOLO model represents a move towards the development of AI-based solutions for the field of hematology to ensure efficiency and accuracy in the analysis of blood samples. Though the Elastic YOLO model has been tested for real-time performance using high-end GPU hardware for the purpose of the evaluation, the testing of the model

for such hardware may be done in the future. This would be a move towards ensuring the availability of a more uniform computer-assisted analysis of blood samples in diverse environments.

Despite the promising results, several limitations should be acknowledged. First, the dataset combines images from multiple sources but remains limited in size compared to large-scale medical imaging datasets. Second, staining protocols and microscope acquisition settings may vary across laboratories, potentially introducing domain shifts when deploying the model in unseen environments. Third, the current evaluation focuses primarily on image-level detection metrics rather than full clinical workflow integration. As highlighted in the failure case analysis, overlapping cells, low contrast regions, and subtle morphological variations can occasionally lead to misclassification between normal and abnormal variants. Future work will therefore focus on cross-institutional validation, domain adaptation strategies, and integration with clinical hematology analyzers to improve robustness across diverse laboratory settings.

Thus, the Elastic YOLO model may be useful in ensuring the availability of a more uniform computer-assisted analysis of blood samples in diverse environments.

## Supporting information

**S1 Fig. Fig is a further set of examples showing how Elastic YOLO works under a range of staining conditions and cell densities.**
(PDF)

**S1 Table. Extends the class-level metrics by including precision, recall, and IoU across a range of IoU thresholds.**
(PDF)

## Acknowledgments

The authors express their sincere gratitude to Dr. K. V. Arulalan, M.D. (Pediatrics), AA Child Care Clinic, Vellore, Tamil Nadu, whose direction in the issue formulation and positive remarks during the writing of this work were of great value to the authors.

The authors further say that they are very thankful to Dr. Radhika, M.D. Pathologist, Government Vellore Medical College, Vellore, Tamil Nadu, and Dr. Mubeen, M.D. Pathologist, Rainbow Hospital, Hyderabad, Telangana, for their professional validation and thorough examination of abnormal or infected cells in peripheral blood smear images. The professionalism of the area played a pivotal role in the diagnostic reliability and clinical significance of the research.

In addition, the authors acknowledge the assistance of Senior Laboratory Technician B. Raghava, Yashoda Hospitals, Hyderabad, whose checking of annotated samples was a major factor for the consistency and accuracy of the ASH Image Bank evaluations. The joint participation of both the pathologists and the senior technician guaranteed that every ASH pathological image used in the study had been independently reviewed and validated with high reliability.

The complete ASH Image Bank samples used in this research are strictly in accordance with the ASH usage and licensing guidelines. The images in this dataset, as well as all the other samples, are completely anonymous and do not contain any identifiable information relating to patients. All of the samples were produced and supplied for educational purposes only and will not be clinically deployed.

The authors would like to acknowledge their strong appreciation for the support, motivation, and guidance that they have received from all contributors that has enabled them to complete this project successfully.

## Author contributions

**Conceptualization:** Neha Margret Issac, Rajakumar K.

**Data curation:** Neha Margret Issac, Rajakumar K.

**Formal analysis:** Neha Margret Issac, Rajakumar K.

**Investigation:** Neha Margret Issac, Rajakumar K.

**Methodology:** Neha Margret Issac, Rajakumar K.

**Project administration:** Rajakumar K.

**Software:** Neha Margret Issac.

**Supervision:** Rajakumar K.

**Validation:** Rajakumar K.

**Visualization:** Rajakumar K.

**Writing – original draft:** Neha Margret Issac, Rajakumar K.

**Writing – review & editing:** Neha Margret Issac, Rajakumar K.

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
