## [Decision Letter · Decision Letter 0]

2 Feb 2026

PONE-D-25-67456Adaptive Elastic Convolution-Based YOLO for Peripheral Blood Smear Cell DetectionPLOS One

Dear Dr. K,

Thank you for submitting your manuscript to PLOS ONE. After careful consideration, we feel that it has merit but does not fully meet PLOS ONE’s publication criteria as it currently stands. Therefore, we invite you to submit a revised version of the manuscript that addresses the points raised during the review process. Please submit your revised manuscript by Mar 19 2026 11:59PM. If you will need more time than this to complete your revisions, please reply to this message or contact the journal office at plosone@plos.org. Please include the following items when submitting your revised manuscript:

We look forward to receiving your revised manuscript.

Kind regards,

Yaseen Ahmed Al-Mulla

Academic Editor

PLOS One

Journal Requirements:

3. Please note that PLOS One has specific guidelines on code sharing for submissions in which author-generated code underpins the findings in the manuscript. In these cases, we expect all author-generated code to be made available without restrictions upon publication of the work.

Please review our guidelines at https://journals.plos.org/plosone/s/materials-and-software-sharing#loc-sharing-code and ensure that your code is shared in a way that follows best practice and facilitates reproducibility and reuse.

5. Please amend your manuscript to include a reference list. References must be placed at the end of the manuscript and numbered in the order that they appear in the text. For more information on the formatting of references, please visit the author guidelines at: http://journals.plos.org/plosone/s/submission-guidelines#loc-reference-style

6. We note you have included a table to which you do not refer in the text of your manuscript. Please ensure that you refer to Table 3 in your text; if accepted, production will need this reference to link the reader to the Table.

8. We are unable to open your Supporting Information file [plos2025.bst, plos_bibtex_sample.bib]. Please kindly revise as necessary and re-upload.

Reviewers' comments:

Reviewer's Responses to Questions

**Comments to the Author**

1. Is the manuscript technically sound, and do the data support the conclusions?

Reviewer #1: Yes

Reviewer #2: Yes

Reviewer #3: Yes

Reviewer #4: Yes

2. Has the statistical analysis been performed appropriately and rigorously? 

Reviewer #1: I Don't Know

Reviewer #2: No

Reviewer #3: Yes

Reviewer #4: Yes

3. Have the authors made all data underlying the findings in their manuscript fully available?

Reviewer #1: Yes

Reviewer #2: Yes

Reviewer #3: No

Reviewer #4: Yes

4. Is the manuscript presented in an intelligible fashion and written in standard English?

Reviewer #1: Yes

Reviewer #2: Yes

Reviewer #3: Yes

Reviewer #4: Yes

5. Review Comments to the Author

Reviewer #1: Elastic YOLO should aim for lightweight, high-precision deployment. The system achieves a throughput of 78 FPS, 30.1 GFLOPs, and 12.4M parameters on a NVIDIA RTX 4090 GPU, occupying approximately 110 MB of storage, demonstrating a favorable balance for real-time analyzers and telemedicine platforms.

The authors acknowledge that current challenges include soft boundaries and overlapping cells. Future work will focus on Explainable AI (XAI), high-resolution processing, and attention modules to enhance clinical credibility and transparency.

Absence of an All-Inclusive Framework: Current models often focus on classifying single cells or specific morphological features, rather than providing a comprehensive framework for multi-class blood cell detection. This limits their applicability to the full spectrum of hematological disorders.

The model occasionally misses small-sized platelets and faint abnormal RBCs, which is attributed to imaging quality and dataset limitations. This indicates that while the model performs well, there are still challenging cases it struggles with.

The dataset used in the study is dominated by a large number of annotations for Red Blood Cells (RBCs)

Reviewer #2: The paper presents Adaptive Elastic Convolution Based YOLO for peripheral blood smear cell detection, I have the following concerns and addressing them can improve the paper:

-Authors need to clarify the conceptual novelty of Elastic YOLO beyond existing deformable convolution and dynamic scaling approaches, as several architectural elements appear incremental and the manuscript does not clearly articulate what fundamentally new capability is introduced.

-Authors need to strengthen the clinical framing by explicitly stating which hematological decisions or diagnostic workflows this model is intended to support, since the current emphasis on detection metrics is not sufficiently connected to clinical relevance.

-Authors need to justify the use of GAN generated abnormal samples more rigorously, including a clear discussion of potential bias and evidence that synthetic data does not artificially inflate performance on rare pathological morphologies.

-Authors need to improve methodological transparency by reporting sensitivity or robustness analyses for key elastic parameters such as depth and width scaling, since current ablation studies show component importance but not stability of design choices.

-Authors need to moderate claims regarding real time and telemedicine readiness by evaluating performance on lower resource hardware and across heterogeneous laboratory settings, as current results rely primarily on high end GPU benchmarks.

Reviewer #3: Confidential Comments to the Editor: PONE-D-25-67456

This manuscript presents a technically sound and comprehensive deep-learning framework (Elastic YOLO) for multiclass detection of peripheral blood smear cells. The methodology is clearly described, results are supported by extensive quantitative analyses, and comparisons with established YOLO variants are appropriate. Overall, the study meets the criteria for technical validity and reproducibility.

The remaining issues relate primarily to clarity, moderation of claims, and transparency rather than fundamental methodological concerns. Statements suggesting near-term clinical use should be moderated to align with the available data, and dataset heterogeneity and reproducibility aspects should be further clarified. These points can be addressed through minor revisions. I therefore recommend Minor Revision.

Reviewer #4: Reviewer Suggestions for PONE-D-25-67456

Title: OK

Abstract:

Define YOLO and EYOLO in the Abstract. I don't know how sophisticated the potential readership for this paper will be, but it would be helpful to do so in the abstract, even though you do define it in the third paragraph of the introduction.

Introduction:

The introductory statements introduce the importance of manual peripheral smear examination. However, this introduction misses the point that modern laboratory practice determines when a manual review is needed and where automated analyzers are weak. Most laboratories use highly automated CBC analyzers that are good at characterizing normal RBCs, WBCs, and platelets. These analyzers flag specific parameters to indicate a need to reflex to manual review.

Automated complete blood count (CBC) analyzers demonstrate excellent accuracy compared to manual methods for most basic parameters, with correlation coefficients typically exceeding 0.97 for white blood cell (WBC), red blood cell (RBC), hemoglobin, and platelet counts.[1-4]

However, accuracy varies considerably for specific differential counts and certain cell populations. For the five-part differential, automated analyzers show good correlation for neutrophils and eosinophils but only fair correlation for lymphocytes and monocytes.[1] Basophil counts remain unreliable across all automated platforms.[1] Compared with 400-cell manual differentials, modern analyzers achieve acceptable accuracy for most leukocyte populations, though performance varies by cell type.[3]

Nucleated red blood cell (NRBC) enumeration shows substantial variation between platforms and compared to microscopy. Newer analyzers like the DxH 800 and XN-2000 outperform older models, with the XN-2000 achieving 90% sensitivity for NRBC detection and maintaining correlation with microscopy up to 15% NRBCs.[3][5] In contrast, some platforms show poor concordance (Kendall's τb = 0.37).[2]

Flagging capabilities for abnormal cells demonstrate moderate sensitivity and specificity. For blast detection, sensitivity ranges from 65% to 76% for most analyzers, though the XN-2000 achieves 97%.[2][5] The XN-2000 also shows greater than 95% sensitivity for abnormal lymphocyte detection, though specificity is only 54%.[5] Importantly, operators cannot rely on blast flagging alone to detect leukemic samples.[1]

1. Comparison of Automated Differential Blood Cell Counts From Abbott Sapphire, Siemens Advia 120, Beckman Coulter DxH 800, and Sysmex XE-2100 in Normal and Pathologic Samples.

American Journal of Clinical Pathology. 2013. Meintker L, Ringwald J, Rauh M, Krause SW.

2. Comparison of Five Automated Hematology Analyzers in a University Hospital Setting: Abbott Cell-Dyn Sapphire, Beckman Coulter DxH 800, Siemens Advia 2120i, Sysmex XE-5000, and Sysmex XN-2000.

Clinical Chemistry and Laboratory Medicine. 2015. Bruegel M, Nagel D, Funk M, et al.

3. Evaluation of the Beckman Coulter UniCel DxH 800, Beckman Coulter LH 780, and Abbott Diagnostics Cell-Dyn Sapphire Hematology Analyzers on Adult Specimens in a Tertiary Care Hospital.

American Journal of Clinical Pathology. 2011. Tan BT, Nava AJ, George TI.

4. A Novel Automated Slide-Based Technology for Visualization, Counting, and Characterization of the Formed Elements of Blood: A Proof of Concept Study.

Archives of Pathology & Laboratory Medicine. 2017. Winkelman JW, Tanasijevic MJ, Zahniser DJ.

5. Performance and Abnormal Cell Flagging Comparisons of Three Automated Blood Cell Counters: Cell-Dyn Sapphire, DxH-800, and XN-2000.

American Journal of Clinical Pathology. 2013. Hotton J, Broothaers J, Swaelens C, Cantinieaux B.

The second paragraph contains an erroneous statement: “RBCs are usually of normal appearance, but during the response to a viral infection, atypical lymphocytes often have a striking resemblance to healthy WBCs [?].”

Atypical lymphocytes do not resemble normal white blood cells (WBCs), particularly normal lymphocytes, in morphology.

• Size and Shape: Atypical lymphocytes are larger than normal lymphocytes, with more abundant cytoplasm that may appear pale or deeply basophilic (blue-staining).

• Nucleus: Their nuclei are irregularly shaped, often indented, lobulated, or grooved, and may contain prominent nucleoli, unlike the round or slightly indented nucleus of a normal lymphocyte.

• Cytoplasmic Features: The cytoplasm frequently molds around surrounding red blood cells, and in some cases, shows radiating basophilia—a distinctive feature.

• Clinical Context: These morphological changes are most commonly seen in reactive conditions, such as infectious mononucleosis (due to Epstein-Barr virus) or other viral infections, but can also be present in certain bacterial infections (e.g., pertussis), drug reactions, or hematological malignancies.

While atypical lymphocytes are a type of WBC, their distinct morphology clearly differentiates them from normal lymphocytes and is a key clue in diagnosing underlying conditions.

Manual microscopy review remains essential in several scenarios: (1) when automated flags indicate blasts, immature granulocytes, or abnormal lymphocytes; (2) for samples with extreme WBC counts (<1.5×10⁹/L or >30×10⁹/L); (3) when platelet counts are <100×10⁹/L or >1000×10⁹/L; (4) for confirmation of any suspected hematologic malignancy; and (5) when clinical suspicion exists despite negative flags.[16-18] Optimized review criteria can reduce manual review rates from 37% to approximately 25-30% while maintaining false-negative rates below 3%.[16][19] The International Consensus Group for Hematology Review provides baseline criteria, but each laboratory should validate and optimize these based on their specific analyzer and patient population.

Automated analyzers exhibit variable performance in detecting abnormal cells across specific clinical scenarios, with significant differences in blast detection in acute leukemia, abnormal lymphocyte flagging in lymphoproliferative disorders, and atypical cell identification in myelodysplastic syndromes.

Blast Detection in Acute Leukemia

For blast detection in acute leukemia, sensitivity varies widely across platforms. The Sysmex XN-2000 achieves the highest sensitivity at 97%, while most other analyzers (Cell-Dyn Sapphire, DxH 800, Advia 2120i, XE-5000) demonstrate 65-76% sensitivity.[1] Importantly, the XN-2000 did not miss any circulating blasts in samples ranging from 0.5% to 95% blasts by microscopy.[2] However, operators cannot rely on blast flagging alone to detect leukemic samples with any analyzer.[3] The Abbott Alinity Hq shows 57.6% overall sensitivity, rising to 70% when limited to samples with ≥5% blasts.[4] In severely leukopenic patients (WBC ≤2.0×10⁹/L), AI-based whole-blood film scanning demonstrates superior performance, detecting all 17 cases with ≥1 blast cell (100% sensitivity) compared with traditional 200-cell counting methods.[5]

Abnormal Lymphocyte Detection in Lymphoproliferative Disorders

For abnormal lymphocyte detection, the Sysmex XN-2000 demonstrates sensitivity greater than 95%, though specificity is only 54%, requiring adaptation of flagging thresholds.[2] Other analyzers show lower performance, with overall flagging efficiency for abnormal WBCs (≥1%) ranging from 72% to 78% across platforms.[6] The XN-2000's WPC channel shows excellent performance for differentiating neoplastic (AUC=0.933) from reactive leukocytosis (AUC=0.900), superior to routine analyzers (AUC=0.630 and 0.635).[7] Cellular population data combined with rapid flow cytometry can help discriminate monoclonal B-cell lymphocytosis from reactive processes, with low lymphocyte volume identifying classical CLL and a B-lymphocyte/total lymphocyte ratio >0.32 suggesting B-malignancy.[8]

Immature Granulocyte Detection

Automated immature granulocyte (IG) enumeration shows good performance with appropriate cutoffs. The XN-2000 and DxH 800 provide useful IG counts with a cutoff <5% and WBC >2,500/mm³.[2] In neonates, automated IG% and absolute IG counts perform equivalently to manual I/T ratios and band counts for identifying infection, with the added advantages of a larger sample size, lower cost, and faster turnaround.[9] The Sysmex DI-60 digital analyzer demonstrates 85.9% sensitivity for detecting immature granulocytes.[10]

Myelodysplastic Syndrome Detection

For MDS, automated analyzers show promise using research parameters beyond routine CBC values. Studies using Beckman Coulter DxH 800 research parameters achieved an ROC/AUC of 0.86 using CBC data alone and 0.93 when combined with molecular data, with 89% sensitivity and 84% specificity.[11] VCS parameters (volume, conductivity, scatter) in neutrophils show significant decreases in MDS patients, with SD-UMALS-NE (standard deviation of upper median angle light scatter in neutrophils) demonstrating 77% sensitivity and 82% specificity.[12] Random forest classifiers using automated CBC parameters achieved an AUC of 0.942 for MDS identification, validated across multiple institutions.[13] However, morphologic examination remains the gold standard for MDS diagnosis, as flow cytometric blast estimation should not substitute for morphology.[14-15]

1. Comparison of Five Automated Hematology Analyzers in a University Hospital Setting: Abbott Cell-Dyn Sapphire, Beckman Coulter DxH 800, Siemens Advia 2120i, Sysmex XE-5000, and Sysmex XN-2000. Clinical Chemistry and Laboratory Medicine. 2015. Bruegel M, Nagel D, Funk M, et al.

2. Performance and Abnormal Cell Flagging Comparisons of Three Automated Blood Cell Counters: Cell-Dyn Sapphire, DxH-800, and XN-2000.

American Journal of Clinical Pathology. 2013. Hotton J, Broothaers J, Swaelens C, Cantinieaux B.

3. Comparison of Automated Differential Blood Cell Counts From Abbott Sapphire, Siemens Advia 120, Beckman Coulter DxH 800, and Sysmex XE-2100 in Normal and Pathologic Samples.

American Journal of Clinical Pathology. 2013. Meintker L, Ringwald J, Rauh M, Krause SW.

4. Multicenter Performance Evaluation of the Abbott Alinity Hq Hematology Analyzer.

Clinical Chemistry and Laboratory Medicine. 2019. Slim CL, Wevers BA, Demmers MWHJ, et al.

5. Advantages of AI-based Whole Blood Film Scanning for Blast Detection in Markedly Leucopenic Blood Films. Annals of Hematology. 2025. Wang G, Zheng L, Fang Z, et al.New

6. Automated Five-Part White Blood Cell Differential Counts. Efficiency of Software-Generated White Blood Cell Suspect Flags of the Hematology Analyzers Sysmex SE-9000, Sysmex NE-8000, and Coulter STKS.

Archives of Pathology & Laboratory Medicine. 1997. Thalhammer-Scherrer R, Knöbl P, Korninger L, Schwarzinger I.

7. Performance of the XN-2000 WPC Channel-Flagging to Differentiate Reactive and Neoplastic Leukocytosis.

Clinical Chemistry and Laboratory Medicine. 2016. Schuff-Werner P, Kohlschein P, Maroz A, et al.

8. Combination of Cellular Population Data and CytoDiff Analyses for the Diagnosis of Lymphocytosis.

Clinical Chemistry and Laboratory Medicine. 2011. Jean A, Boutet C, Lenormand B, et al.

9. Comparing Automated vs Manual Leukocyte Differential Counts for Quantifying the 'Left Shift' in the Blood of Neonates.

Journal of Perinatology: Official Journal of the California Perinatal Association. 2016. MacQueen BC, Christensen RD, Yoder BA, et al.

10. Red and White Blood Cell Morphology Characterization and Hands-on Time Analysis by the Digital Cell Imaging Analyzer DI-60.

PloS One. 2022. Kweon OJ, Lim YK, Lee MK, Kim HR.

11. Beyond the Routine CBC: Machine Learning and Statistical Analyses Identify Research CBC Parameter Associations With Myelodysplastic Syndromes and Specific Underlying Pathogenic Variants.

Journal of Clinical Pathology. 2023. Pozdnyakova O, Niculescu RS, Kroll T, et al.

12. Automated Leukocyte Parameters Are Useful in the Assessment of Myelodysplastic Syndromes.

Cytometry. Part B, Clinical Cytometry. 2021. Shestakova A, Nael A, Nora V, Rezk S, Zhao X.

13. Automated Screening for Myelodysplastic Syndromes Through Analysis of Complete Blood Count and Cell Population Data Parameters.

American Journal of Hematology. 2014. Raess PW, van de Geijn GJ, Njo TL, et al.

14. Myelodysplastic Syndromes.

National Comprehensive Cancer Network. Updated 2026-01-12.

15. Diagnosis and Classification of Myelodysplastic Syndromes.

Blood. 2023. Hasserjian RP, Germing U, Malcovati L.

16. Validation and Optimization of Criteria for Manual Smear Review Following Automated Blood Cell Analysis in a Large University Hospital.

Archives of Pathology & Laboratory Medicine. 2013. Pratumvinit B, Wongkrajang P, Reesukumal K, Klinbua C, Niamjoy P.

17. Laboratory Productivity and the Rate of Manual Peripheral Blood Smear Review: A College of American Pathologists Q-Probes Study of 95,141 Complete Blood Count Determinations Performed in 263 Institutions.

Archives of Pathology & Laboratory Medicine. 2006. Novis DA, Walsh M, Wilkinson D, St Louis M, Ben-Ezra J.

18. Elimination of Instrument-Driven Reflex Manual Differential Leukocyte Counts. Optimization of Manual Blood Smear Review Criteria in a High-Volume Automated Hematology Laboratory.

American Journal of Clinical Pathology. 2003. Lantis KL, Harris RJ, Davis G, Renner N, Finn WG.

19. Development of the Personalized Criteria for Microscopic Review Following Four Different Series of Hematology Analyzer in a Chinese Large-Scale Hospital.

Chinese Medical Journal. 2010. Cui W, Wu W, Wang X, et al.

20. The Rate of Manual Peripheral Blood Smear Reviews in Outpatients.

Clinical Chemistry and Laboratory Medicine. 2009. Froom P, Havis R, Barak M.

Related Work, YOLOv5–Based Studies, Advancements in YOLOv8 and Its Derived Architectures, Transformer-Integrated & Morphology-Aware Architectures, Identified Challenges in Current Research, Proposed Research Goals and Strategic Objectives :

This suggestion might be unnecessary depending on the audience, but it would be helpful to define the following terms for those unfamiliar with them:

DL-Based (Deep Learning)

Darknet Framework (open-source neural network framework)

PyTorch (open-source deep learning library)

Vision Transformers (deep learning models that apply the Transformer architecture—initially designed for natural language processing—to computer vision tasks)

Affine Transformations (geometric transformations that preserve collinearity (points on a line remain on a line) and ratios of distances between points on a line)

Otherwise, these sections are fine.

Proposed Methodology: Good

Dataset, Annotations, and Preprocessing:

The complexity of WBC abnormalities is minimized in this description. Need more granularity. The interest is in identifying WBC cell-types that automated systems struggle with, such as monocytes and basophils. Has EYOLO improved this detection?

Dataset Structure: This section raises a few concerns. As I outlined previously, existing automated systems do well with normal RBCs, WBCs, and platelets. They have difficulty with low cell counts, basophils, monocytes, NRBCs, and blasts. You state that there are only a few abnormalities in the dataset. Please clarify. Detecting standard cell types is of minimal interest. How does this improve detection of abnormals?

Train–Validation–Test Split: Good process

YOLOv5 Baseline Framework, Elastic YOLO Architecture Adaptation, Architectural Comparisons from a Comparative Perspective, Detection Logic and Visualization: Very clear

Performance Evaluation Metrics, Loss Function Formulation, Model Complexity Analysis, Algorithmic Representation and Complexity Evaluation: Unambiguous and appropriate to the hypothesis.

Results and Discussion:

It is of greater interest to understand the performance of this method improvement in identifying analytic challenges posed by NRBCs, leukemic blasts, lymphomas, viral atypical lymphocytes, and in identifying cell types such as basophils and monocytes. That data is not presented and might not be available in your analysis, but it is of paramount clinical interest to determine how functional this technical improvement might be. You have clearly demonstrated the superiority of YOLOv8 over YOLOv5 if that was the central hypothesis. Still, that alone does not indicate how it would be deployed or how it would improve on the existing challenges in our automated systems and manual peripheral blood examinations. Your discussion should address these questions to some extent, even if the data and results do not yet indicate a direction.

6. PLOS authors have the option to publish the peer review history of their article (what does this mean?). If published, this will include your full peer review and any attached files.

Reviewer #1: **Yes:** Dr Zil a Rubab

Reviewer #2: No

Reviewer #3: No

Reviewer #4: No

---

## [Author Response · Author response to Decision Letter 1]

14 Feb 2026

The detailed, point-by-point responses to the reviewers’ comments have been carefully prepared and are included in the file titled “Revised Manuscript with Track Changes.pdf.”

We sincerely thank the Editor and the reviewers for their time, effort, and constructive feedback, which have greatly helped us improve the quality, clarity, and technical rigor of the manuscript.

---

## [Decision Letter · Decision Letter 1]

8 Mar 2026

PONE-D-25-67456R1Adaptive Elastic Convolution-Based YOLO for Peripheral Blood Smear Cell DetectionPLOS One

Dear Dr. K,

Thank you for submitting your manuscript to PLOS ONE. After careful consideration, we feel that it has merit but does not fully meet PLOS ONE’s publication criteria as it currently stands. Therefore, we invite you to submit a revised version of the manuscript that addresses the points raised during the review process.

We look forward to receiving your revised manuscript.

Kind regards,

Yaseen Ahmed Al-Mulla

Academic Editor

PLOS One

**Journal Requirements:**

Reviewers' comments:

Reviewer's Responses to Questions

**Comments to the Author**

1. If the authors have adequately addressed your comments raised in a previous round of review and you feel that this manuscript is now acceptable for publication, you may indicate that here to bypass the “Comments to the Author” section, enter your conflict of interest statement in the “Confidential to Editor” section, and submit your "Accept" recommendation.

Reviewer #2: All comments have been addressed

Reviewer #3: (No Response)

Reviewer #4: All comments have been addressed

2. Is the manuscript technically sound, and do the data support the conclusions?

Reviewer #2: (No Response)

Reviewer #3: Yes

Reviewer #4: (No Response)

3. Has the statistical analysis been performed appropriately and rigorously? 

Reviewer #2: (No Response)

Reviewer #3: Yes

Reviewer #4: (No Response)

4. Have the authors made all data underlying the findings in their manuscript fully available?

Reviewer #2: (No Response)

Reviewer #3: Yes

Reviewer #4: (No Response)

5. Is the manuscript presented in an intelligible fashion and written in standard English?

Reviewer #2: (No Response)

Reviewer #3: Yes

Reviewer #4: (No Response)

6. Review Comments to the Author

Reviewer #2: (No Response)

Reviewer #3: Minor Revisions for Manuscript: PONE-D-25-67456_R1

Abstract, and summary

Bring main key performance numbers and baselines earlier in the abstract.

Reduce overlap between abstract and Author Summary; add one concrete clinical use‑case like telemedicine triage.

Introduction

Shorten disease lists by grouping conditions and emphasize the core gap like morphology, rare cells, and manual PBS burden.

Briefly state specific limitations of prior YOLO/CNN PBS work like overlap, subtle variants, stain/domain shifts, etc.

Data and acronyms clarity

Summarize datasets: names, sizes, classes, sources, stain/imaging conditions, and clinician annotation process.

Ensure all symbols and acronyms are defined once and used consistently.

Metrics definitions

Define mAP, AP, IoU, FPS, FPN, PANet, CIoU, etc., at their first use.

Ensure consistent notation for Cpos, Cneg, Mpos, Mneg and clearly explain F1, precision, and recall.

Results, limitations, and style

Report per‑class AP, highlighting abnormal classes (ARBC, AWBC, APL), plus a confusion matrix or brief misclassification table.

Expand limitations like dataset size/diversity, centers, stains, lack of external cohort, focus on image‑level metrics, and link future work to observed failure cases.

Tighten long sentences, fix minor grammar, and standardize terminology (PBS, EYOLO). Also ensure Data Availability, Ethics, and Competing Interests in the text match the submission forms and that references follow journal style.

Reviewer #4: (No Response)

7. PLOS authors have the option to publish the peer review history of their article (what does this mean?). If published, this will include your full peer review and any attached files.

Reviewer #2: No

Reviewer #3: No

Reviewer #4: No

---

## [Author Response · Author response to Decision Letter 2]

11 Mar 2026

We thank the Editor and the Reviewers for their valuable comments and constructive suggestions. All the comments have been carefully addressed, and the manuscript has been revised accordingly. A detailed, point-by-point response to the reviewer comments is provided in the separate file titled “Response to Reviewers.pdf.”

---

## [Decision Letter · Decision Letter 2]

29 Mar 2026

Adaptive Elastic Convolution-Based YOLO for Peripheral Blood Smear Cell Detection

PONE-D-25-67456R2

Dear Dr. K,

We’re pleased to inform you that your manuscript has been judged scientifically suitable for publication and will be formally accepted for publication once it meets all outstanding technical requirements.

Kind regards,

Yaseen Ahmed Al-Mulla

Academic Editor

PLOS One

Additional Editor Comments (optional):

Reviewers' comments:

Reviewer's Responses to Questions

**Comments to the Author**

1. If the authors have adequately addressed your comments raised in a previous round of review and you feel that this manuscript is now acceptable for publication, you may indicate that here to bypass the “Comments to the Author” section, enter your conflict of interest statement in the “Confidential to Editor” section, and submit your "Accept" recommendation.

Reviewer #3: All comments have been addressed

2. Is the manuscript technically sound, and do the data support the conclusions?

Reviewer #3: Yes

3. Has the statistical analysis been performed appropriately and rigorously? 

Reviewer #3: Yes

4. Have the authors made all data underlying the findings in their manuscript fully available?

Reviewer #3: Yes

5. Is the manuscript presented in an intelligible fashion and written in standard English?

Reviewer #3: Yes

6. Review Comments to the Author

Reviewer #3: Comments to Authors PONE-D-25-67456_R2

The authors have adequately addressed the previous reviewer comments and improved the clarity and presentation of the manuscript. The study is now well-organized, and the results are clearly presented. I have no further major concerns and consider the manuscript suitable for publication in its current form.

7. PLOS authors have the option to publish the peer review history of their article (what does this mean?). If published, this will include your full peer review and any attached files.

Reviewer #3: No

---

## [Editor Report · Acceptance letter]

PONE-D-25-67456R2

PLOS One

Dear Dr. K,

I'm pleased to inform you that your manuscript has been deemed suitable for publication in PLOS One. Congratulations! Your manuscript is now being handed over to our production team.

Kind regards,

on behalf of

Dr. Yaseen Ahmed Al-Mulla

Academic Editor

PLOS One